# SPECTRAL ANNEALING FOR SCALABLE ISING OPTIMIZATION

## ABSTRACT

Large-scale Ising optimization underlies critical optimization and learning applications including correlation clustering, MAP inference, and energy-based models. Existing methods face a fundamental efficiency-quality tradeoff: spectral relaxations achieve polynomial-time complexity but deliver approximate solutions with significant optimality gaps, while metaheuristics approach optimal solutions but scale poorly to large instances. The core limitation of spectral methods is their single-shot nature—they solve exactly one eigenvalue problem, providing no solution exploration and exhibiting acute sensitivity to normalization choices. We introduce *spectral annealing* (SpecAnn), a new spectral paradigm that transforms spectral methods from single-shot approximations into systematic exploration-based optimization. Our key innovation is to explore a continuous parameterized family of spectral relaxations $\mathbf{N}_\alpha = \mathbf{D}_\alpha^{-1/2}\mathbf{A}\mathbf{D}_\alpha^{-1/2}$ for $\alpha \in [0, 1]$, interpolating from raw adjacency to full normalization, and harvest solutions across this spectrum. We introduce a *diagonal predictor* that exploits the algebraic structure of this continuation path for efficient traversal, combined with GPU-accelerated batched operations that leverage massive parallelism across both eigenvalue computations and solution refinement. Experiments on problems up to 8.4 million variables demonstrate that SpecAnn achieves near optimal solutions with runtimes of 1–57 seconds, bridging fast but inaccurate spectral methods and slow but high-quality metaheuristics. SpecAnn delivers $83\times$ speedup over recent GPU-accelerated methods while maintaining superior solution quality, and scales successfully beyond 262K variables where traditional metaheuristics fail.

## 1 INTRODUCTION

Modern large-scale optimization increasingly demands solutions to large nonlinear problems under strict real-time constraints. 5G LDPC decoders process codewords with $10^4+$ spins in sub-millisecond latency (Liu et al., 2024), city-scale traffic systems formulate signal timing as QUBO instances with tens of thousands of binary variables that must be resolved within seconds (Salloum et al., 2025; Inoue et al., 2021), and GPU-accelerated portfolio optimization handles $\sim10^5$ scenarios requiring continuous re-solving as market data streams in (Huo et al., 2025). These applications expose a fundamental tension in existing methods. Spectral relaxation approaches satisfy the computational requirements through polynomial-time eigenvalue formulations but deliver approximate solutions with significant optimality gaps (Delorme & Poljak, 1993a; Trevisan, 2012). Conversely, metaheuristic methods including simulated annealing and parallel tempering can achieve near-optimal solutions but require extensive sampling that scales poorly beyond moderate problem sizes (Kirkpatrick et al., 1983; Swendsen & Wang, 1986; Lucas, 2014). Recent GPU-accelerated approaches such as Quasi-Quantum Annealing (Ichikawa & Arai, 2024) and discrete Langevin methods (Sun et al., 2023) demonstrate improved scalability yet still rely on gradient-based or sampling dynamics that demand careful convergence tuning. We introduce Spectral annealing (SpecAnn) to bridge this gap by transforming spectral methods from single-shot approximations into systematic parameter space exploration, achieving high solution quality while scaling to problems with of millions of variables through GPU parallelism.

**The Ising optimization landscape.** The Ising model minimizes $\min_{\mathbf{s} \in \{-1,+1\}^n} E(\mathbf{s}) = -\frac{1}{2}\mathbf{s}^\top\mathbf{As}$ over symmetric coupling matrices $\mathbf{A} \in \mathbb{R}^{n \times n}$, appearing ubiquitously in combinatorial optimization and machine learning from correlation clustering (Bansal et al., 2004a; Kunegis et al., 2010a) and MAP inference (Boykov et al., 2001; Wainwright & Jordan, 2008) to energy-based models (Hopfield, 1982; Ackley et al., 1985). Exact methods including commercial solvers achieve global optima but become prohibitive beyond $n \lesssim 10^3$ due to exponential scaling. Classical metaheuristics such as simulated annealing (Kirkpatrick et al., 1983) and parallel tempering (Swendsen & Wang, 1986) scale to larger instances but suffer from slow convergence requiring hundreds of thousands of iterations. Quantum and quantum-inspired approaches (Kadowaki & Nishimori, 1998; Johnson et al., 2011) face connectivity constraints or parameter sensitivity.

Spectral relaxation offers an attractive middle ground by replacing discrete optimization with continuous eigenvalue problems, achieving $O(\text{nnz}(\mathbf{A}) \cdot I)$ complexity where $I$ denotes eigenvalue iterations and $\text{nnz}(\mathbf{A})$ counts nonzero elements. These methods replace binary constraints with $\|\mathbf{s}\|_2^2 = n$, solve for the principal eigenvector of a spectral matrix $\mathbf{M}$ derived from $\mathbf{A}$, and round to obtain feasible solutions. However, classical spectral approaches suffer from a critical limitation: *they solve exactly one eigenvalue problem and stop*. This single-shot paradigm provides no solution exploration, exhibits acute parameter sensitivity, and wastes computational potential when eigensolvers converge rapidly. For signed graphs in Ising problems, parameter sensitivity becomes especially severe—the adjacency matrix $\mathbf{A}$ preserves signal strength but suffers from degree imbalance, while the normalized Laplacian $\mathbf{L}^{\pm} = \mathbf{I} - \mathbf{D}^{-1/2}\mathbf{A}\mathbf{D}^{-1/2}$ achieves degree balancing but may over-normalize (Kunegis et al., 2010b). The optimal choice depends on instance structure in ways that resist principled a priori selection, as Figure 1 demonstrates.

**From single-shot approximation to systematic exploration.** Rather than committing to one spectral formulation, we ask: what if spectral methods could systematically explore a continuous family of relaxations while preserving their computational efficiency? This represents a paradigm shift from viewing spectral methods as fixed approximations to recognizing them as controllable exploratory tools. The key insight is that spectral relaxations form a rich continuous family—systematic traversal with efficient warm-starting unlocks exploration at minimal additional cost, transforming the fundamental nature of spectral optimization.

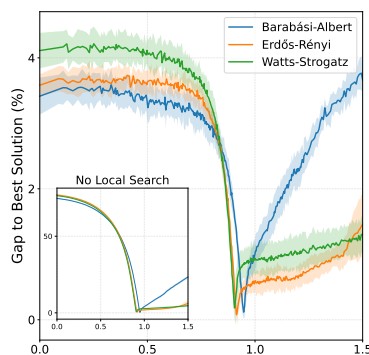

**Our contribution: Spectral Annealing.** We introduce SpecAnn, which achieves this transformation through three integrated components:

*(1) Algorithmic framework.* Rather than solving one spectral relaxation, we parameterize normalization via $\alpha \in [0, 1]$, defining $\mathbf{N}_\alpha = \mathbf{D}_\alpha^{-1/2}\mathbf{A}\mathbf{D}_\alpha^{-1/2}$ where $\mathbf{D}_\alpha = \text{diag}(\mathbf{d}^\alpha)$ interpolates from raw adjacency ($\alpha = 0$) to full normalization ($\alpha = 1$). Solving eigenvalue problems along an annealing schedule $\{\alpha_k\}_{k=0}^K$ and harvesting best solutions across all normalizations provides 2–5× quality improvements over single-parameter spectral methods while maintaining $O(\text{nnz}(\mathbf{A}))$ scaling per step.

Figure 1: Energy gaps across $\alpha$ values on 65K-node instances. Optimal $\alpha$ varies significantly across instances with performance differences exceeding 10%, motivating systematic exploration over fixed choices.

*(2) Computational innovation.* Our *diagonal predictor* exploits the algebraic structure of $\mathbf{N}_\alpha$ to enable efficient warm-starting of successive eigenvalue computations. This reduces Lanczos iterations by 25–40% compared to cold starts, providing 35% overall speedup. We provide rigorous eigen-drift analysis quantifying spectral evolution along the annealing path and introduce stability-preserving spectral shifts using Gershgorin's theorem. Combined with GPU-accelerated batched operations, this achieves near-linear scaling to problems with 8M+ variables.

*(3) Theoretical foundation.* We establish strong inapproximability results for general Ising optimization and derive *instance-dependent universal energy bounds* as byproducts of the spectral continuation framework. These bounds bracket the true optimum—the spectral bound always lies above the true energy while below the relaxed estimate—providing interpretable optimality certificates without requiring ground truth.

**Empirical validation.** Comprehensive experiments across 66 Gset benchmark instances, real-world signed networks (up to 131K nodes), and synthetic graphs (up to 8.4M nodes) demonstrate that SpecAnn uniquely occupies a favorable balanced regime: 0–5% energy gaps with 1–57 second runtimes across all tested scales. On 2M-node problems, SpecAnn achieves 0.0% gap in 56.9 seconds versus QQA's 6.55% gap in 4733 seconds—representing 83× speedup with superior quality. At 262K nodes, SpecAnn matches simulated annealing's 0.35% gap quality while delivering 884× speedup (6.7 versus 5901 seconds). Critically, SpecAnn scales beyond 262K nodes where traditional metaheuristics fail, successfully optimizing problems with millions of variables where exact methods become intractable. Ablation studies confirm that the diagonal predictor contributes measurable efficiency gains (35% speedup) while the two-component design (spectral initialization + local search) provides 2–5% quality improvement over either component alone.

**Related work.** SpecAnn shares conceptual ties with continuation methods in optimization. Simulated annealing (Kirkpatrick et al., 1983) varies temperature stochastically, while resolution parameters in community detection (Reichardt & Bornholdt, 2006) reveal multiscale structure. Our approach deterministically explores normalization parameters for signed graphs where degree definitions require careful handling.

Classical spectral methods include the Delorme-Poljak eigenvalue bound (Delorme & Poljak, 1993a;b) and Trevisan's algorithm (Trevisan, 2012), achieving 0.531 approximation for max-cut, though these focus on unsigned graphs. For signed networks, Kunegis et al. (Kunegis et al., 2010b) introduced the signed Laplacian, while SPONGE (Cucuringu et al., 2019) optimizes positive-to-negative edge weight ratios. However, these signed network studies do not optimize the Ising energy, but a ratio-based signed Laplacian objective. Sulem et al. (Sulem et al., 2021) introduce regularization for sparse graphs, however, the approach uses fixed normalizations rather than a systematic exploration.

Beyond spectral methods, recent GPU-accelerated approaches optimize continuous relaxations. QQA (Ichikawa & Arai, 2024) performs gradient-based updates with entropy annealing and parallel-run communication, while iSCO (Sun et al., 2023) simulates discrete Langevin dynamics with first-order approximations and multi-bit Metropolis updates. Unlike these sampling-driven relaxations, SpecAnn follows a gradient-free and deterministic eigenvector continuation along $\mathbf{N}_\alpha$, exploiting spectral structure through normalization homotopy rather than gradient or sampling dynamics.

The Goemans-Williamson SDP (Goemans & Williamson, 1995) provides 0.878 approximation for max-cut but lacks guarantees for general Ising models. SDP methods achieve strong MaxCut guarantees (Goemans & Williamson, 1995; Burer & Monteiro, 2003; Yurtsever et al., 2021) but are specialized for non-negative weights. SpecAnn directly handles general Ising problems with signed couplings, scaling to 8.4M variables on problems beyond SDP applicability.

Our diagonal predictor connects to homotopy continuation (Li & Rhee, 1989) and path-following methods (Allgower & Georg, 1993), but introduces $\alpha$-parameterization specifically for Ising optimization with rigorous drift control and systematic normalization exploration for signed graphs.

## 2 BACKGROUND: SPECTRAL METHODS FOR ISING OPTIMIZATION

### 2.1 ISING MODEL AND APPLICATIONS

**The Ising Model and Ground State Problem.** The Ising model, originally introduced to study ferromagnetism in statistical physics (Ising, 1925), defines an energy function over binary spin variables $\mathbf{s} \in \{-1, +1\}^n$:

$$E(\mathbf{s}) = - \sum_{(i,j) \in \mathcal{E}} J_{ij} s_i s_j - \sum_{i=1}^{n} h_i s_i + C, \tag{1}$$

where $J_{ij}$ represents pairwise interactions between spins, $h_i$ are external magnetic fields, and $C$ is an additive constant. At finite temperature $T > 0$, the system follows a Boltzmann distribution $P(\mathbf{s}) \propto \exp(-E(\mathbf{s})/T)$, with configurations of lower energy having higher probability.

The *ground state* corresponds to the spin configuration that minimizes this energy functional. We refer to this optimization problem as *Min-Ising*:

$$\mathbf{s}^* = \arg \min_{\mathbf{s} \in \{\pm 1\}^n} E(\mathbf{s}), \tag{2}$$

which is NP-hard in general (Barahona, 1982). An equivalent formulation is the Quadratic Unconstrained Binary Optimization (QUBO) problem $\min_{\mathbf{x} \in \{0,1\}^n} \mathbf{x}^T Q \mathbf{x}$, obtained via the change of variables $s_i = 2x_i - 1$.

**Universality and Combinatorial Applications.** Many combinatorial optimization problems admit natural Ising formulations. The MaxCut problem on a weighted graph $G = (V, \mathcal{E}, \mathbf{w})$, which seeks to partition vertices to maximize the total weight of edges crossing the partition, corresponds to Ising interactions $J_{ij} = -w_{ij}$ with external fields $h_i = 0$. The energy becomes

$$E(\mathbf{s}) = \sum_{(i,j) \in \mathcal{E}} w_{ij} s_i s_j, \tag{3}$$

where opposite spins ($s_i \neq s_j$) contribute $-w_{ij}$ and aligned spins contribute $+w_{ij}$. Thus, minimizing $E(\mathbf{s})$ maximizes the cut value. Other problems naturally expressible as Ising instances include graph partitioning, vertex cover, independent set, number partitioning, and constraint satisfaction problems (Lucas, 2014), making Ising optimization a canonical testbed for combinatorial solvers Chen et al. (2023); Ichikawa & Arai (2024).

**Applications in Machine Learning.** Ising models appear throughout machine learning: Markov Random Fields reduce to Ising MAP inference (Koller & Friedman, 2009), feature selection encodes as QUBO (Nguyen & Welch, 2013), correlation clustering maps to MaxCut on signed graphs (Bansal et al., 2004b), and modularity maximization

reformulates as MaxCut variants (Newman, 2006). These applications motivate scalable solvers handling hundreds of thousands of variables.

**Computational Complexity and Hardness.** MaxCut, where all interaction weights are nonnegative ($J_{ij} \leq 0$), is NP-hard (Karp, 1972) even on restricted graph classes (Garey & Johnson, 1979). The best known approximation is the Goemans-Williamson SDP algorithm (Goemans & Williamson, 1995), achieving ratio $\alpha_{\text{GW}} \approx 0.878$, which is optimal under the Unique Games Conjecture (Khot, 2002; Khot et al., 2007).

For general Ising instances with arbitrary interaction weights, we establish strong inapproximability by reduction from modularity clustering Dinh et al. (2015).

**Theorem 1** (Hardness of approximation for Min-Ising). *Let $E_{\text{opt}} = \min_{\mathbf{s} \in \{\pm 1\}^n} E(\mathbf{s})$ denote the ground state energy. For any constant $\alpha \geq 1$, an $\alpha$-approximation algorithm outputs $\mathbf{s}_{\text{alg}}$ satisfying $E(\mathbf{s}_{\text{alg}}) \leq \alpha \cdot E_{\text{opt}}$. Unless $\mathrm{P} = \mathrm{NP}$, no polynomial-time $\alpha$-approximation exists for any finite constant $\alpha \geq 1$.*

This demonstrates that the ferromagnetic structure of MaxCut is essential for approximation guarantees. For general Ising models with both ferromagnetic ($J_{ij} > 0$) and antiferromagnetic ($J_{ij} < 0$) interactions, no constant-factor approximation is possible in polynomial time unless P=NP. The proof is in Appendix A.1.

These hardness results imply that practical algorithms must trade off quality guarantees against computational efficiency. While SDP-based methods achieve near-optimal quality on MaxCut, their $O(n^{3.5})$ complexity and $O(n^2)$ memory limit scalability. Conversely, local search heuristics scale to millions of variables but provide no guarantees and often converge to poor local optima. This work addresses the gap by developing spectral methods that maintain computational efficiency while systematically exploring the solution space for the general Ising models.

## 2.2 Spectral Relaxation Framework

We minimize the Ising energy $E(\mathbf{s}) = -\frac{1}{2}\mathbf{s}^\top \mathbf{A}\mathbf{s} - \sum_i h_i s_i$ over binary spins $\mathbf{s} \in \{-1, +1\}^n$, where $\mathbf{A} \in \mathbb{R}^{n \times n}$ is symmetric and $\mathbf{h} \in \mathbb{R}^n$ represents local fields. Local fields are eliminated by introducing auxiliary spin $s_{n+1} = +1$ and setting $\tilde{A}_{i,n+1} = h_i$, yielding field-free form with $A_{ii} = 0$.

The challenge stems from discrete constraints over exponentially large solution spaces. Coupling matrices in ML contexts exhibit extreme sparsity, heterogeneous degree distributions (Mercado et al., 2019), and complex sign patterns mixing ferromagnetic and antiferromagnetic interactions—making instances simultaneously too large for exact methods and too structured for generic metaheuristics.

Spectral methods replace discrete constraints with continuous relaxation $\|\mathbf{s}\|_2^2 = n$, yielding bound $E(\mathbf{s}) \geq -\frac{1}{2}\lambda_1(\mathbf{M}) \cdot$ scale factor for matrix $\mathbf{M}$ derived from $\mathbf{A}$. The principal eigenvector $\mathbf{v}_1$ is computed and rounded via $\hat{\mathbf{s}} = \text{sign}(\mathbf{v}_1)$. This achieves $O(\text{nnz}(\mathbf{A}) \cdot I)$ complexity with $I \ll n$ eigenvalue iterations, maps naturally to GPU parallelization with minimal hyperparameter tuning, making a promising candidate for a robust and scalable solver.

## 2.3 The Normalization Dilemma

The critical design choice is the spectral matrix $\mathbf{M}$, which determines both bound quality and solution structure. For signed graphs, this presents fundamental trade-offs.

**Raw adjacency** ($\mathbf{M} = \mathbf{A}$) yields bound $E(\mathbf{s}) \geq -\frac{1}{2}\lambda_1(\mathbf{A})n$ and preserves full signal strength, but suffers from degree imbalance where high-degree vertices dominate $\mathbf{v}_1$ regardless of optimal spin arrangements.

**Normalized Laplacian** addresses this using absolute degrees $d_i = \sum_j |A_{ij}|$ (necessary since naive degrees can vanish for signed graphs (Kunegis et al., 2010b)). The signed normalized Laplacian $\mathbf{L}^{\pm} = \mathbf{I} - \mathbf{D}^{-1/2}\mathbf{A}\mathbf{D}^{-1/2}$ with $\mathbf{D} = \text{diag}(\mathbf{d})$ balances vertex influence but may over-normalize when degree heterogeneity carries optimization signal.

Neither approach dominates universally. The optimal choice depends on instance-specific properties that resist principled a priori selection, causing classical spectral methods to underperform their computational potential(Cucuringu et al., 2019; Sulem et al., 2021). This motivates systematic exploration across normalization choices rather than single-formulation commitment.

# 3 Spectral Annealing: Systematic Parameter Exploration

This section introduces our systematic framework for exploring the spectral parameter space that transforms single-shot spectral methods into exploration-based optimization while maintaining computational efficiency.

## 3.1 The $\alpha$-Parameterized Spectral Family

Rather than committing to a single spectral matrix, we define a continuous family of spectral relaxations that smoothly interpolates between raw adjacency and full normalization. This parameterization enables systematic exploration of the trade-off between signal preservation and degree balancing without requiring a priori knowledge of optimal settings for each instance.

**Family definition.** For $\alpha \in [0, 1]$, define the $\alpha$-*degree matrix* $\mathbf{D}_\alpha = \mathrm{diag}(\mathbf{d}^\alpha)$ where $\mathbf{d}^\alpha$ denotes element-wise exponentiation of the absolute degree vector $\mathbf{d}$ with $d_i = \sum_j |A_{ij}|$. The $\alpha$-*normalized adjacency matrix* is

$$\mathbf{N}_\alpha = \mathbf{D}_\alpha^{-1/2} \mathbf{A} \mathbf{D}_\alpha^{-1/2}$$

This parameterization smoothly interpolates between two extremes: $\mathbf{N}_0 = \mathbf{A}$ (raw adjacency preserving full signal strength) and $\mathbf{N}_1 = \mathbf{D}^{-1/2} \mathbf{A} \mathbf{D}^{-1/2}$ (full normalization achieving degree balance). Intermediate values $\alpha \in (0, 1)$ provide hybrid normalizations that partially balance degrees while retaining signal strength proportional to $d_i^{\alpha-1}$.

**Exploration strategy.** Our SpecAnn approach solves a sequence of eigenvalue problems along a schedule $\mathcal{S} = \{\alpha_k \in [0, 1] : k = 0, \ldots, K\}$. For each $\alpha_k \in \mathcal{S}$:

1. Compute the principal eigenpair $(\lambda_k, \mathbf{y}_k)$ of $\mathbf{N}_{\alpha_k}$
2. Round to spins: $\hat{\mathbf{s}}(\alpha_k) = \mathrm{sign}(\mathbf{D}_{\alpha_k}^{-1/2} \mathbf{y}_k)$
3. Apply local search improvement
4. Track the best energy across all $\alpha_k$ values

This systematic exploration harvests solutions from multiple points in the spectral parameter space, adaptively discovering which normalizations work best for each instance without requiring domain expertise.

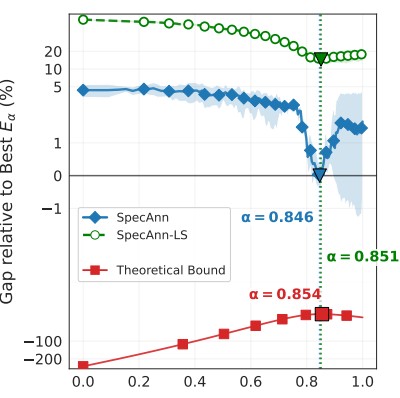

Figure 2: Relative gaps to the ground state energy (black line at 0) for SpecAnn, SpecAnn-LS (SpecAnn without local search), and universal energy bound derived in Eq. 5 (bottom red curve) for BA graphs ($N = 1024$).

**Universal energy bounds.** The parameterization yields valid energy bounds for every $\alpha$ simultaneously. For any Ising configuration $\mathbf{s} \in \{-1, +1\}^n$, setting $\mathbf{z} = \mathbf{D}_\alpha^{1/2} \mathbf{s}$ and applying the Rayleigh-Ritz theorem gives $\mathbf{s}^\top \mathbf{A} \mathbf{s} = \mathbf{z}^\top \mathbf{N}_\alpha \mathbf{z} \leq \lambda_1(\alpha) \|\mathbf{z}\|_2^2 = \lambda_1(\alpha) \sum_{i=1}^n d_i^\alpha$, where $\lambda_1(\alpha)$ denotes the largest eigenvalue of $\mathbf{N}_\alpha$. This yields the energy bound

$$E(\mathbf{s}) \geq \max_{\alpha \in [0,1]} -\frac{1}{2} \lambda_1(\alpha) \sum_i d_i^\alpha, \tag{5}$$

separating a *spectral shape* term $\lambda_1(\alpha)$ from a *degree scale* term $\sum_i d_i^\alpha$.

The universal energy bound enables an *instance-specific approximation ratio* by comparing the energy of obtained solutions with a universal lower bound. Figure 2 shows that this bound—maximized at $\alpha \in [0.8, 1]$, surpasses previous spectral bounds achieved at $\alpha = 0$ and $\alpha = 1$ and a clear correlation between tighter lower bounds and solvers reaching their minimal energy gap. Additional experiments and analysis for our new bound are provided in Appendix C.5.

## 3.2 Efficient Annealing via GPU-accelerated Warm-Starting

The primary computational challenge in spectral exploration is that solving $K$ independent eigenvalue problems could increase costs by a factor of $K$, potentially negating the efficiency advantages of spectral methods. Our key algorithmic breakthrough is a *diagonal predictor* that exploits the smooth algebraic structure of the $\alpha$-parameterized family to enable efficient warm-starting, dramatically reducing the computational cost per annealing step.

The fundamental observation enabling efficient warm-starting is that eigenvectors exhibit approximate invariance under diagonal rescaling. Specifically, if $\mathbf{y}(\alpha)$ is the principal eigenvector of $\mathbf{N}_\alpha$, then the *unscaled vector* $\mathbf{x}(\alpha) = \mathbf{D}_\alpha^{-1/2}\mathbf{y}(\alpha)$ changes more slowly with $\alpha$ than $\mathbf{y}(\alpha)$ itself. This suggests that a good predictor for $\mathbf{y}(\alpha + \Delta)$ can be obtained by appropriately rescaling $\mathbf{y}(\alpha)$.

**Diagonal predictor.** For a small step $\Delta > 0$, we predict the next eigenvector using

$$\mathbf{y}_{\text{pred}}(\alpha + \Delta) = \frac{\mathbf{D}_{\alpha+\Delta}^{1/2}\mathbf{D}_\alpha^{-1/2}\mathbf{y}(\alpha)}{\|\mathbf{D}_{\alpha+\Delta}^{1/2}\mathbf{D}_\alpha^{-1/2}\mathbf{y}(\alpha)\|_2} = \frac{\mathbf{D}^{\Delta/2}\mathbf{y}(\alpha)}{\|\mathbf{D}^{\Delta/2}\mathbf{y}(\alpha)\|_2} \tag{6}$$

This predictor requires only $O(n)$ operations (diagonal scaling and normalization) and exploits the congruence relation $\mathbf{N}_{\alpha'} = \mathbf{D}^{-(\alpha'-\alpha)/2}\mathbf{N}_\alpha\mathbf{D}^{-(\alpha'-\alpha)/2}$ that governs the evolution of the spectral family. The predictor provides a high-quality initialization for iterative eigenvalue solvers (Lanczos, LOBPCG), reducing the number of iterations.

**Eigenvalue prediction.** We also predict the corresponding eigenvalue using the Rayleigh quotient:

$$\lambda_{\text{pred}}(\alpha + \Delta) = \mathbf{y}(\alpha)^\top\mathbf{N}_{\alpha+\Delta}\mathbf{y}(\alpha) = (\mathbf{D}^{-\Delta/2}\mathbf{y}(\alpha))^\top\mathbf{N}_\alpha(\mathbf{D}^{-\Delta/2}\mathbf{y}(\alpha)) \tag{7}$$

This prediction is first-order accurate in $\Delta$ and becomes exact when degrees are uniform ($\mathbf{D} = c\mathbf{I}$), providing both warm-start vectors and eigenvalue estimates for convergence monitoring.

**Computational impact.** The diagonal predictor transforms spectral exploration from prohibitively expensive to computationally feasible. While solving $K$ independent eigenvalue problems would require $O(K \cdot \text{nnz}(\mathbf{A}) \cdot I_{\text{cold}})$ operations, warm-starting reduces this to $O(K \cdot \text{nnz}(\mathbf{A}) \cdot I_{\text{warm}} + K \cdot n)$ where typically $I_{\text{warm}} \ll I_{\text{cold}}$. For dense annealing schedules, the amortized cost per step approaches that of computing matrix-vector products, making exploration nearly as efficient as single-shot methods.

### 3.2.1 GPU IMPLEMENTATION

The efficiency of SpecAnn depends critically on the ability to solve large sparse eigenvalue problems and refine multiple candidate solutions at scale. Our implementation leverages GPU-accelerated sparse linear algebra to achieve both goals.

**Eigenvalue computation.** We use an iterative Krylov subspace solver specialized for real symmetric matrices, implemented in operator form. Rather than explicitly forming the normalized matrix $\mathbf{N}_\alpha = \mathbf{D}_\alpha^{-1/2}\mathbf{A}\mathbf{D}_\alpha^{-1/2}$ or the shifted matrix $\mathbf{S}_\rho(\alpha) = \rho\mathbf{I} - \mathbf{N}_\alpha$, the solver only evaluates their action on a vector:

$$\mathbf{v} \mapsto \rho\mathbf{v} - \mathbf{D}_\alpha^{-1/2}\big(\mathbf{A}(\mathbf{D}_\alpha^{-1/2}\mathbf{v})\big).$$

This reduces each iteration to two diagonal scalings in $O(n)$ time and one sparse matrix–vector product in $O(\text{nnz}(\mathbf{A}))$ time, using $\mathbf{A}$ stored in CSR format. Although $\mathbf{A}$ itself is explicitly stored, the operator is matrix-free with respect to $\mathbf{S}_\rho(\alpha)$ and avoids constructing dense matrices. Warm-starting across successive $\alpha$ values reuses the previous leading eigenvector as initialization, substantially reducing the number of iterations required.

**Batched local search with asynchronous update.** To improve discrete solutions obtained from spectral rounding, we apply a GPU-accelerated local search. Multiple candidate solutions are stacked into a flat tensor and processed in batches, with warps evaluating spin updates across solutions concurrently. Such a batch update effectively exploit the power of GPUs in parallel computation. For each candidate spin configuration $s \in \{-1, +1\}^n$, the energy change $\Delta_i = s_i \sum_j A_{ij}s_j$ is computed, and spins are flipped *asynchronously* whenever the energy change is negative. This asynchronous update provides a higher degree of the parallelization and solution exploration, due to the uncertainty in the parallel spin flips. However, deterministically evaluating and flipping all improving spins can lead to synchronous oscillations and cycling, especially under massively parallel asynchronous updates. To mitigate this effect, we introduce stochasticity by randomly selecting only a subset of vertices (90% by default) at each iteration. As a consequence, some improving flips are intentionally skipped, which breaks deterministic update patterns and reduces the risk of cycling while preserving high GPU parallel efficiency.

Persistent operator objects are maintained across annealing steps by updating only diagonal scalings and the shift $\rho$, avoiding repeated setup costs. All core operations—sparse matrix–vector multiplication, diagonal scaling, normalization, and dot products—are mapped to GPU primitives (cuSPARSE and custom kernels). The combination of operator-form eigensolving, warm-starting, and batched local refinement yields significant runtime improvements, as confirmed by ablation studies in Section 4.

### 3.3 THEORETICAL GUARANTEES FOR STABLE EXPLORATION

To ensure that SpecAnn provides reliable and predictable behavior, we establish rigorous theoretical foundations governing the evolution of eigenvalues and eigenvectors along the annealing path. These results provide explicit step-size control and guarantee stable convergence properties.

**Spectral continuity and drift bounds.** The algebraic structure of the $\alpha$-parameterized family enables precise analysis of how spectral properties evolve. Define $\mathbf{\Gamma} = \mathrm{diag}(\log \mathbf{d})$ where the logarithm is taken element-wise. The following result quantifies eigenvalue sensitivity:

**Theorem 2** (Eigenvalue drift bound). *For any $\alpha, \alpha' \in [0,1]$ with $\Delta = \alpha' - \alpha$, each eigenvalue $\lambda_k(\alpha)$ of $\mathbf{N}_\alpha$ satisfies*

$$|\lambda_k(\alpha') - \lambda_k(\alpha)| \leq |\Delta| \|\mathbf{\Gamma}\|_2 \sup_{t \in [\alpha, \alpha']} \|\mathbf{N}_t\|_2 \tag{8}$$

The proof follows from the exact derivative $\frac{d}{d\alpha}\mathbf{N}_\alpha = -\frac{1}{2}(\mathbf{\Gamma}\mathbf{N}_\alpha + \mathbf{N}_\alpha \mathbf{\Gamma})$ and standard perturbation theory arguments. The bound shows that eigenvalue drift is controlled by the step size $|\Delta|$, degree heterogeneity $\|\mathbf{\Gamma}\|_2 = \max_i |\log d_i|$, and spectral norm $\|\mathbf{N}_\alpha\|_2$.

**Eigenvector stability.** For eigenvector perturbations, we apply the Davis-Kahan $\sin\Theta$ theorem:

**Theorem 3** (Eigenvector drift along the $\alpha$-path). *Let $A \in \mathbb{R}^{n \times n}$ be symmetric, let $d_i = \sum_j |A_{ij}|$ and*

$$D_\alpha = \mathrm{diag}(d_i^\alpha), \qquad N_\alpha = D_\alpha^{-1/2} A D_\alpha^{-1/2}, \qquad \alpha \in [0,1].$$

*Let $\lambda_1(\alpha) \geq \lambda_2(\alpha) \geq \cdots \geq \lambda_n(\alpha)$ be the eigenvalues of $N_\alpha$, and let $y_1(\alpha)$ be a unit principal eigenvector of $N_\alpha$. Define $\Gamma = \mathrm{diag}(\log d_i)$, $\gamma(\alpha) = \lambda_1(\alpha) - \lambda_2(\alpha)$.*

*Fix an interval $[a,b] \subseteq [0,1]$ and assume that $\gamma(\alpha) > 0$ for all $\alpha \in [a,b]$. Then for any $\alpha, \alpha' \in [a,b]$ we have*

$$\sin\angle\big(y_1(\alpha), y_1(\alpha')\big) \leq \frac{|\alpha' - \alpha|\, \|\Gamma\|_2 \sup_{t \in [a,b]} \|N_t\|_2}{\inf_{t \in [a,b]} \gamma(t)}.$$

This result provides explicit control over eigenvector rotation, ensuring that small annealing steps lead to small changes in the principal eigenvector direction. The theorem 3 applies on intervals where the largest eigenvalue is simple, i.e., $\gamma(\alpha) > 0$. At isolated degeneracies with $\lambda_1(\alpha) = \lambda_2(\alpha)$, a single principal eigenvector is not uniquely defined and may rotate rapidly. Algorithmically, this does not require a full restart: we still warm start across $\alpha$ but may need more iterations near such points, and the Davis–Kahan bound does not apply there. In unsigned (or gauge-balanced) connected graphs, Perron–Frobenius guarantees a simple largest eigenvalue for all $\alpha$, so $\gamma(\alpha) > 0$ on $[0,1]$.

**Adaptive step-size control.** The drift bounds immediately suggest a principled step-size rule. To maintain eigenvector stability with principal angle changes bounded by a tolerance $\tau \in (0,1)$, we enforce

$$|\Delta| \leq \tau \frac{\gamma(\alpha_k)}{\|\mathbf{\Gamma}\|_2 \|\mathbf{N}_{\alpha_k}\|_2} \tag{9}$$

This adaptive rule automatically reduces step sizes when degree heterogeneity is high ($\|\mathbf{\Gamma}\|_2$ large) or spectral gaps are small ($\gamma(\alpha_k)$ small), ensuring stable convergence while maximizing annealing efficiency.

**Positive definiteness via spectral shift.** To ensure robust numerical behavior, we introduce a spectral shift parameter $\rho > 0$ and solve the equivalent positive definite problem

$$\mathbf{S}_\rho(\alpha) = \rho\mathbf{I} - \mathbf{N}_\alpha \succ 0 \tag{10}$$

The shifted matrix has the same eigenvectors as $\mathbf{N}_\alpha$ but eigenvalues $\nu_k = \rho - \lambda_k$, making the largest eigenvalue of $\mathbf{N}_\alpha$ correspond to the smallest eigenvalue of $\mathbf{S}_\rho(\alpha)$. Using Gershgorin's circle theorem, we have a sufficient condition:

**Theorem 4** (Gershgorin-safe positive definiteness). *If $\rho > \max_i d_i^{1-\alpha}$, then $\mathbf{S}_\rho(\alpha) \succ 0$.*

The combination of drift analysis, adaptive step-size control, and spectral shifting provides a complete theoretical foundation for reliable spectral exploration, ensuring that our method exhibits predictable and stable behavior across diverse problem instances.

**Complexity.** Each annealing step involves: (i) spectral shift in $O(n)$ (degrees precomputed once), (ii) diagonal rescaling in $O(n)$, (iii) eigenvalue computation in $O(\text{nnz}(\mathbf{A}) \cdot I_k)$ with $I_k$ Krylov iterations, (iv) rounding in $O(n)$, and (v) local search in $O(L \cdot \text{nnz}(\mathbf{A}))$ for $L$ passes. The eigenvalue solve dominates, making iteration count the key factor. Warm-starting lowers $I_k$ by reusing eigenvectors across $\alpha$, while deduplication reduces local search cost by skipping repeated solutions. Their effectiveness depends on spectral smoothness and solution diversity. For $K$ steps, the overall cost is $O\left(K \cdot \text{nnz}(\mathbf{A}) \cdot \bar{I} + L \cdot K^* \cdot \frac{\text{nnz}(\mathbf{A})}{P_{\text{eff}}}\right)$, where $\bar{I}$ is the average warm-started iteration count, $K^* \leq K$ is the number of unique solutions, and $P_{\text{eff}}$ the GPU parallel efficiency. More analysis on the number of iterations is shown in the Appendix A.7.

**Memory.** The space complexity is $O(\text{nnz}(\mathbf{A}) + mn)$, storing $\mathbf{A}$ in CSR plus $m$ Krylov vectors ($m \ll n$), scaling linearly with problem size.

## 4 EXPERIMENTAL EVALUATION

We conduct comprehensive experiments to evaluate SpecAnn time-quality trade-offs for large-scale Ising optimization in comparison to the state-of-the-art Ising solvers.

### 4.1 EXPERIMENTAL SETUP

The experimental evaluation compares SpecAnn against established baselines across three dataset categories: (1) *Real-world signed networks* (Leskovec et al., 2010): Correlation clustering on three signed social networks (Epinions, Slashdot, Wikipedia election datasets) with 7K–131K nodes. (2) *Gset benchmark* (Ye, 2003): 66 connected instances with 800–40,000 vertices spanning diverse graph structures. (3) *Synthetic graphs*: Barabási-Albert (scale-free) and Erdős-Rényi (random) graphs ranging from $n = 64$ to $n = 8,388,608$ vertices with edge weights in $[-100, 100]$.

Baseline methods include exact optimization via Gurobi (Gurobi Optimization, LLC, 2023) (2-hour limit), meta-heuristics (Simulated Annealing (Kirkpatrick et al., 1983), Simulated Bifurcation (Goto et al., 2019), Quasi-Quantum Annealing (Ichikawa & Arai, 2024), Tabu Search (Glover, 1989), Parallel Tempering (Earl & Deem, 2005)), greedy construction with 1000 restarts, and spectral relaxations using adjacency (SpecAdj) and normalized signed Laplacian (SpecSLap) matrices with identical post-processing refinement as SpecAnn. All baseline methods were systematically tuned on BA instances ($n = 2^6$ to $2^{20}$) with 2-hour time limits to ensure fair comparison (detailed tuning protocol in Appendix C.1). SA used 100 samples with 1000 sweeps per trajectory (balancing quality and $10\times$ speedup over 10000 sweeps). SB employed heated discrete dynamics with 1024 agents. QQA used batch size 100, diversity parameter 0.2, background field range $[-3.0, 0.1]$, and 3000 epochs following Ichikawa & Arai (2024) with configurations optimizing quality-runtime tradeoffs. All experiments were conducted on Intel Xeon Platinum 8276L CPUs (256GB RAM) and NVIDIA A100 (PCIe, 40 GB).

**Evaluation metrics.** We measure two key metrics: (1) *Energy gap to best solution*: the relative percentage gap between a method's solution energy and the best solution energy found across all methods on that instance, i.e., $100 \times (E_{\text{method}} - E_{\text{best}})/|E_{\text{best}}|$. (2) *Time-to-target*: the time required for a method to reach or surpass a reference solution quality (we use SpecAnn's quality as the target), with methods failing to reach the target within one hour marked as unsuccessful.

### 4.2 RESULTS

**MaxCut and correlation clustering.** Table 1 presents performance comparisons across the Gset benchmark (66 MaxCut instances, excluding the disconnected instances) and three signed social networks for correlation clustering. The results reveal a fundamental time-quality tradeoff across method categories. At the quality-focused end, Simulated Annealing (SA) achieves the best solution quality with 0.1% average gap on Gset and optimal solutions (0.0% gap) on all signed networks, followed closely by Simulated Bifurcation (SB) with 1.1% gap on Gset. However, both methods require substantial computational time: SA ranges from 7.2 seconds on Gset to over 1200 seconds on large networks (Epinions and Slashdot), while SB requires 13 seconds on Gset. Additionally, five traditional metaheuristics (Gurobi, Greedy, TS, PT, SB) exhibit fundamental scalability limitations, completely failing on the two largest signed networks (Epinions: 131K nodes, Slashdot: 82K nodes) due to time or memory constraints. Gurobi achieves 2.1% gap on Gset but requires an average of 2223 seconds per instance, rendering it impractical for routine optimization.

Table 1: Performance on MaxCut (Gset, 66 instances) and correlation clustering (signed networks). "–" indicates timeout/memory limit.

| | MaxCut | | Correlation Clustering | | | | | |
| | Time (s) | Gap(%) | Time (s) | | | Gap (%) | | |
| Method | Gset | Gset | Epinions | Slashdot | WikiElec | Epinions | Slashdot | WikiElec |
|---|---|---|---|---|---|---|---|---|
| Gurobi | 2222.8 | 2.1 | – | – | 128.1 | – | – | 0.0 |
| Greedy | 951.1 | 15.0 | – | – | 2299.1 | – | – | 0.1 |
| TS | 549.3 | 18.2 | – | – | 2021.7 | – | – | 98.7 |
| PT | 491.2 | 79.8 | – | – | 1447.2 | – | – | 35.5 |
| SA | 7.2 | 0.1 | 1210.1 | 744.1 | 25.4 | 0.0 | 0.0 | 0.0 |
| SB | 12.6 | 1.1 | – | – | 29.5 | – | – | 0.0 |
| QQA | 106.1 | 1.1 | 65.3 | 39.7 | 7.2 | 20.0 | 22.2 | 6.7 |
| SpecAdj | 0.1 | 22.3 | 0.1 | 0.0 | 0.0 | 0.3 | 0.7 | 0.1 |
| SpecSLap | 0.1 | 10.7 | 0.1 | 0.0 | 0.0 | 2.9 | 3.5 | 13.2 |
| SpecAnn | 2.2 | 4.6 | 1.9 | 1.8 | 1.7 | 0.2 | 0.5 | 0.0 |

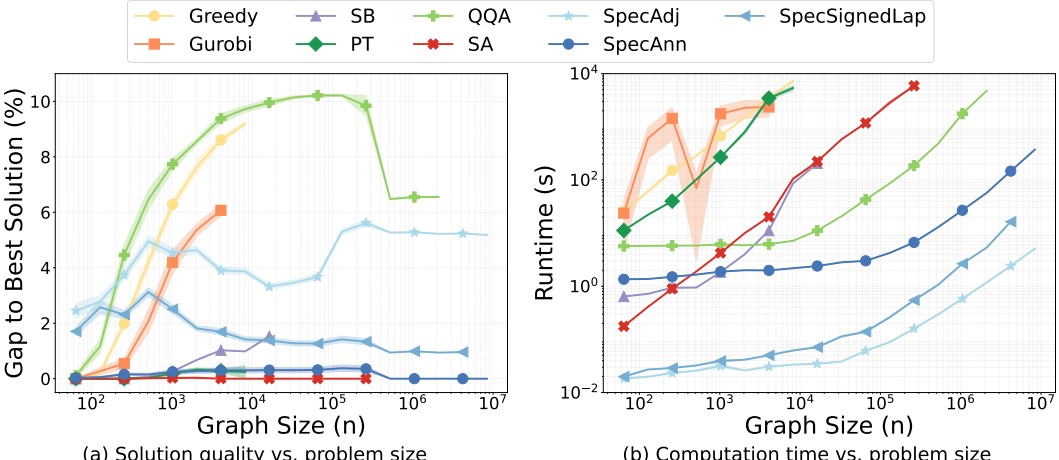

(a) Solution quality vs. problem size   (b) Computation time vs. problem size

Figure 3: Scalability results on Barabási-Albert (BA) networks. SpecAnn maintains consistent solution quality while scaling efficiently, outperforming metaheuristics on large instances. Due to the poor performance and near-vertical scaling of TS, with accuracy reaching 96.32% on the largest graph size of 8,192, TS results were excluded from the figure for clarity.

At the speed-focused end, pure spectral methods (SpecAdj and SpecSLap) demonstrate minimal computational overhead of 0.1 seconds or less across all instances. However, this efficiency comes at the cost of substantial quality degradation: SpecAdj suffers 22.3% gap on Gset, while SpecSLap exhibits high variability on correlation clustering instances with gaps ranging from 2.9% to 13.2%. These methods provide rapid approximate solutions but lack the refinement necessary for high-quality optimization.

SpecAnn occupies a balanced intermediate position. On Gset, it achieves 4.6% gap in 2.2 seconds, substantially better than pure spectral methods with modest overhead. On signed networks, SpecAnn demonstrates 0.0–0.5% gaps in 1.7–1.9 seconds, approaching SA's quality while requiring 13–640× less time. Ablation studies confirm the diagonal predictor reduces Lanczos iterations by 25–40%, providing 35% speedup (Table 2). Disabling local search increases gaps from 0.0–0.4% to 5–17%, while random initialization with local search underperforms SpecAnn by 2–5% (Figure 6), confirming both components are essential. This positions SpecAnn as achieving near-metaheuristic quality at near-spectral speed, *filling the performance gap between rapid but inaccurate spectral relaxations and slow but high-quality iterative methods.*

**Analysis of QQA.** QQA represents recent GPU-accelerated Ising optimization. On Gset, QQA achieves 1.1% gap in 106 seconds, matching SB's quality but requiring 8× longer runtime. While QQA scales to all correlation clustering instances where traditional metaheuristics fail, it exhibits substantially higher gaps versus SpecAnn: 20.0% versus 0.2% on Epinions, 22.2% versus 0.5% on Slashdot. SpecAnn demonstrates 22–34× faster execution on these networks

(1.8–1.9 seconds versus 40–65 seconds). On 2M-node synthetic graphs, SpecAnn achieves $83\times$ speedup over QQA. Time-to-target analysis (Tables 4, 5) shows SA requires sub-second convergence for small instances but over 2400 seconds at $n = 524{,}288$ versus SpecAnn's 13 seconds, while QQA fails to reach SpecAnn's quality on dense BA instances.

**Synthetic graphs.** Figure 3 illustrates scalability on Barabási-Albert networks from 64 to 8.4M nodes. SA achieves optimal solutions (0.0% gap) up to 262K nodes but requires 5901 seconds at this scale, while traditional metaheuristics (Greedy, Gurobi, PT, SB) fail beyond $n \leq 16{,}384$. Pure spectral methods scale efficiently (0.02–5.31 seconds for 2M nodes) but suffer 1–6% gaps. QQA scales to 2M nodes but exhibits 6.45–10.22% gaps with prohibitive runtimes (4733 seconds at $n = 2{,}097{,}152$).

SpecAnn uniquely combines quality and efficiency. It maintains 0.0–0.4% gaps across all scales with near-linear runtime growth (1.35–56.9 seconds for 64–2M nodes), achieving optimal 0.0% solutions on the largest instances (524K–2M nodes) where SA fails. At $n = 262{,}144$, SpecAnn achieves 0.35% gap with $884\times$ speedup over SA (6.7 versus 5901 seconds). At $n = 2{,}097{,}152$, SpecAnn demonstrates $83\times$ speedup over QQA (56.9 versus 4733 seconds) while achieving superior quality (0.0% versus 6.55% gap). Similar scaling is observed on Erdős-Rényi graphs (Figure 4).

**Performance trade-offs.** Results reveal a three-way tradeoff: quality-focused methods (SA, SB) achieve near-optimal solutions but fail beyond 262K nodes; speed-focused spectral methods execute in about 6 seconds with 1–6% gaps; QQA scales to 2M nodes with 6–10% gaps requiring thousands of seconds. SpecAnn achieves 0.0–0.4% gaps in 1–57 seconds across all scales to 8.4M nodes, delivering $83–884\times$ speedups with superior or competitive quality.

### 4.3   ABLATION STUDY

We validate SpecAnn's components on Barabási-Albert graphs ($n = 64$ to 8M). **Diagonal predictor.** Table 2 compares three strategies: SpecAnn`[-WS]` (no warm-start), SpecAnn (diagonal predictor), and SpecAnn`[Sec]` (secant predictor). The diagonal predictor reduces Lanczos iterations by 25–40% (Appendix Table 10), achieving 35% speedup (3.84s versus 5.19s) with identical quality (0.04% gap). The secant predictor provides marginal improvement (3.67s) with diminishing returns.

Table 2: Impact of warm-start strategies on solution quality and runtime (averaged across BA instances).

| Variant | Gap (%) | Time (s) |
|---|---|---|
| SpecAnn`[-WS]` | 0.04 | 5.19 |
| SpecAnn | 0.04 | 3.84 |
| SpecAnn`[Sec]` | 0.05 | 3.67 |

**Two-component design.** Figure 6 validates both components. SpecAnn`[-LS]` (no local search) suffers 5–17% gaps versus SpecAnn's 0.0–0.4%. Random initialization with local search (LS-only) underperforms SpecAnn by 2–5%, confirming spectral initialization provides crucial high-quality starting points. Both components are essential.

## 5   CONCLUSION

We introduced spectral annealing, which transforms spectral methods from single-shot approximations into systematic exploration-based optimization by traversing a continuous $\alpha$-parameterized family of spectral relaxations. The diagonal predictor exploits algebraic structure for efficient warm-starting, enabling rapid traversal of the parameter space. Experiments on problems up to 8.4M variables demonstrate 0–5% energy gaps with 1–57 second runtimes. SpecAnn achieves $83\times$ speedup over QQA on 2M-node problems (0.0% versus 6.55% gap) and scales successfully beyond 262K nodes where traditional metaheuristics fail. Ablation studies confirm the effectiveness of our diagonal predictor and the two-component design (spectral initialization and local search) delivers 2–5% quality improvement over either component alone. SpecAnn operates directly on QUBO/Ising formulations, requiring only standard eigenvalue solvers and local search, enabling immediate deployment across applications.

**Limitation**: SpecAnn's effectiveness depends on degree of heterogeneity. On regular or near-regular graphs (e.g., some Gset instances), the $\alpha$-parameterized family $\mathbf{N}_\alpha = \mathbf{D}_\alpha^{-1/2} \mathbf{A} \mathbf{D}_\alpha^{-1/2}$ exhibits limited variation across $\alpha$ values, reducing exploration benefits. Future work will investigate alternative parameterizations beyond degree-based normalization, such as regularization-based paths or spectral gap-guided trajectories.

### REPRODUCIBILITY

The code/data is available at `https://anonymous.4open.science/r/SpectralAnnealing-0DFC/`.

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

## A  APPENDIX: THEORETICAL PROOFS

This appendix provides complete proofs for the theoretical results establishing the rigorous foundations of SpecAnn. The results are organized to build from basic spectral properties to the complete drift analysis that enables stable annealing schedules. We begin with a summary of the used notations in Table 3.

| Symbol | Meaning |
|---|---|
| $\mathbf{A} \in \mathbb{R}^{n \times n}$ | symmetric signed adjacency, zero diagonal |
| nnz($\mathbf{A}$) | number of nonzero entries in $\mathbf{A}$ (sparsity measure) |
| $d_i = \sum_j |A_{ij}|, \mathbf{D} = \mathrm{diag}(\mathbf{d})$ | absolute degrees, degree matrix |
| $\mathbf{D}_\alpha = \mathrm{diag}(\mathbf{d}^\alpha)$ | degree scaling along path |
| $\mathbf{N}_\alpha = \mathbf{D}_\alpha^{-1/2} \mathbf{A} \mathbf{D}_\alpha^{-1/2}$ | $\alpha$-normalized adjacency |
| $\mathbf{L}_\alpha^\pm = I - \mathbf{N}_\alpha$ | signed normalized Laplacian |
| $\lambda_k(\alpha), \gamma(\alpha)$ | eigenvalues of $\mathbf{N}_\alpha$, spectral gap |
| $\mathbf{S}_\rho(\alpha) = \rho I - \mathbf{N}_\alpha$ | shifted SPD form |
| $\mathbf{Q}_\rho(\alpha) = \rho \mathbf{D}_\alpha - \mathbf{A}$ | diagonally scaled SPD form |
| $\mathbf{\Gamma} = \mathrm{diag}(\log \mathbf{d})$ | log-degree diagonal |
| $E(\mathbf{s}) = -\frac{1}{2} \mathbf{s}^\top \mathbf{A} \mathbf{s}$ | Ising energy |

Table 3: Notations

## A.1 HARDNESS OF APPROXIMATION FOR MIN-ISING

We present the proof of Theorem 1 by a gap-preserving reduction to the modularity maximization problem Agarwal & Kempe (2008). We begin with the introduction of the modularity maximization problem.

**Modularity maximization and its inapproximability.** We recall the standard modularity maximization problem. Let $G = (V, E)$ be an undirected graph with $m = |E|$ edges, adjacency matrix $A = (A_{ij})$, degrees $k_i = \sum_j A_{ij}$, and modularity matrix

$$B_{ij} = A_{ij} - \frac{k_i k_j}{2m}.$$

For a bipartition encoded by $s \in \{\pm 1\}^{|V|}$, the Newman–Girvan modularity Agarwal & Kempe (2008) is

$$\max_{s \in \{\pm 1\}^{|V|}} Q(s) = \frac{1}{4m} s^\top B s = \frac{1}{4m} \sum_{i,j} B_{ij} s_i s_j,$$

and we denote its optimal value by $Q_{\mathrm{OPT}}$.

**Theorem 5** (Inapproximability of modularity maximization Dinh et al. (2015)). *For every constant $\rho > 0$, there is no polynomial-time algorithm that, on every input graph with optimal modularity $Q_{OPT} > 0$, outputs $s$ with*

$$Q(s) \geq \rho \, Q_{OPT},$$

*unless* $\mathrm{P} = \mathrm{NP}$.

**Proof of Theorem 1.** Our proof is a simple reduction from modularity maximization to Min-Ising followed by applying Theorem 5.

**Modularity maximization as a Min-Ising instance.** Given a graph $G$ as above, we construct a Min-Ising instance on the same vertex set by setting all local fields to zero, $h_i = 0$ for all $i$, and defining pairwise couplings $J_{ij} = J_{ji} = \frac{B_{ij}}{2m}$ for $i < j$, and choosing the constant term $C = -\frac{1}{4m} \sum_{i=1}^n B_{ii}$. Using symmetry of $B$, we have

$$Q(s) = \frac{1}{4m} \sum_{i,j} B_{ij} s_i s_j = \frac{1}{4m} \Big( \sum_i B_{ii} + 2 \sum_{i<j} B_{ij} s_i s_j \Big) = \frac{1}{2m} \sum_{i<j} B_{ij} s_i s_j + \frac{1}{4m} \sum_i B_{ii}.$$

Therefore, for any $s \in \{\pm 1\}^n$,

$$
\begin{aligned}
H(s) &= -\sum_{i<j} J_{ij} s_i s_j + C \\
&= -\frac{1}{2m} \sum_{i<j} B_{ij} s_i s_j - \frac{1}{4m} \sum_i B_{ii} \\
&= -\frac{1}{4m} \Big( \sum_i B_{ii} + 2 \sum_{i<j} B_{ij} s_i s_j \Big) = -Q(s).
\end{aligned}
$$

Hence minimizing $H$ is exactly equivalent to maximizing modularity:

$$\text{OPT}_{\text{Ising}} = \min_s H(s) = \min_s \big(-Q(s)\big) = -\max_s Q(s) = -Q_{\text{OPT}}.$$

In particular, for all instances with $Q_{\text{OPT}} > 0$ (the regime of Theorem 5), we have

$$\text{OPT}_{\text{Ising}} = -Q_{\text{OPT}} < 0.$$

**Min-Ising Inapproximability.** Assume, for contradiction, that there exists a polynomial-time $\alpha$-approximation algorithm $\mathcal{A}$ for Min-Ising, for some fixed constant $0 < \alpha \leq 1$. Apply $\mathcal{A}$ to the Ising instance $H(s) = -Q(s)$ constructed above, and let $s_{\text{alg}}$ be its output. For instances with $Q_{\text{OPT}} > 0$, we have $\text{OPT}_{\text{Ising}} = -Q_{\text{OPT}} < 0$, we have,

$$H(s_{\text{alg}}) \ \leq \ \alpha \,\text{OPT}_{\text{Ising}} = \alpha \,(-Q_{\text{OPT}}).$$

Using $H(s) = -Q(s)$, this becomes

$$-Q(s_{\text{alg}}) \ \leq \ -\alpha \, Q_{\text{OPT}},$$

and multiplying by $-1$ flips the inequality:

$$Q(s_{\text{alg}}) \ \geq \ \alpha \, Q_{\text{OPT}}.$$

Thus $\mathcal{A}$ induces a polynomial-time multiplicative $\alpha$-approximation algorithm for modularity maximization, contradicting Theorem 5 for any fixed $\alpha > 0$.

Therefore, no polynomial-time $\alpha$-approximation algorithm for Min-Ising exists for any finite constant $0 < \alpha \leq 1$, unless $P = NP$.

## A.2 SPECTRAL SHIFT AND POSITIVE DEFINITENESS

We first establish the fundamental result ensuring that our spectral shift produces well-conditioned positive definite problems.

**Theorem 6** (Gershgorin-Safe Positive Definiteness). *Let $\mathbf{Q}_\rho(\alpha) = \rho \mathbf{D}_\alpha - \mathbf{A}$ where $\mathbf{A}$ is symmetric with $A_{ii} = 0$ and $\mathbf{D}_\alpha = diag(\mathbf{d}^\alpha)$ with $d_i = \sum_j |A_{ij}|$. If $\rho > \max_i d_i^{1-\alpha}$, then $\mathbf{Q}_\rho(\alpha) \succ 0$ and hence $\mathbf{S}_\rho(\alpha) = \rho \mathbf{I} - \mathbf{N}_\alpha \succ 0$.*

*Proof.* We apply Gershgorin's circle theorem to establish strict diagonal dominance of $\mathbf{Q}_\rho(\alpha)$. For each row $i$, the diagonal entry is

$$(\mathbf{Q}_\rho)_{ii} = \rho d_i^\alpha \tag{11}$$

and the off-diagonal sum is

$$\sum_{j \neq i} |(\mathbf{Q}_\rho)_{ij}| = \sum_{j \neq i} |A_{ij}| = d_i \tag{12}$$

since $A_{ii} = 0$ by assumption.

The condition $\rho > \max_i d_i^{1-\alpha}$ implies $\rho > d_i^{1-\alpha}$ for all $i$, which gives

$$\rho d_i^\alpha > d_i^{1-\alpha} d_i^\alpha = d_i \tag{13}$$

Therefore, $(\mathbf{Q}_\rho)_{ii} > \sum_{j \neq i} |(\mathbf{Q}_\rho)_{ij}|$ for all $i$, establishing strict diagonal dominance.

By Gershgorin's circle theorem, all eigenvalues of $\mathbf{Q}_\rho(\alpha)$ lie in the union of discs

$$\mathcal{D}_i = \{z \in \mathbb{C} : |z - \rho d_i^\alpha| \leq d_i\} \tag{14}$$

Since $\rho d_i^\alpha > d_i \geq 0$, each disc satisfies $\text{Re}(z) \geq \rho d_i^\alpha - d_i > 0$ for all $z \in \mathcal{D}_i$. Consequently, all eigenvalues have positive real parts. For a real symmetric matrix, all eigenvalues are real, hence all eigenvalues are positive, implying $\mathbf{Q}_\rho(\alpha) \succ 0$.

The relation $\mathbf{S}_\rho(\alpha) = \mathbf{D}_\alpha^{-1/2} \mathbf{Q}_\rho(\alpha) \mathbf{D}_\alpha^{-1/2}$ is a congruence transformation with the nonsingular matrix $\mathbf{D}_\alpha^{1/2}$, which preserves positive definiteness. Therefore, $\mathbf{S}_\rho(\alpha) \succ 0$. $\square$

## A.3 SPECTRAL CONTINUITY AND ENVELOPE PROPERTIES

The following results establish how spectral properties evolve smoothly along the annealing path and provide uniform bounds on the spectral family.

**Theorem 7** (Monotone Norm Envelope). *Let $d_{\min} = \min_i d_i$ and $d_{\max} = \max_i d_i$. For any $\alpha, \alpha' \in [0, 1]$ with $\Delta = \alpha' - \alpha$,*

$$d_{\max}^{-|\Delta|} \|\mathbf{N}_\alpha\|_2 \leq \|\mathbf{N}_{\alpha'}\|_2 \leq d_{\min}^{-|\Delta|} \|\mathbf{N}_\alpha\|_2 \tag{15}$$

*In particular, if $d_{\min} \geq 1$ and $\Delta \geq 0$, the mapping $\alpha \mapsto \|\mathbf{N}_\alpha\|_2$ is non-increasing.*

*Proof.* We use the exact congruence relation $\mathbf{N}_{\alpha'} = \mathbf{R}\mathbf{N}_\alpha\mathbf{R}$ where $\mathbf{R} = \mathbf{D}^{-\Delta/2}$ is diagonal.

For the upper bound, since $\mathbf{R}$ is diagonal,

$$\|\mathbf{N}_{\alpha'}\|_2 = \|\mathbf{R}\mathbf{N}_\alpha\mathbf{R}\|_2 \leq \|\mathbf{R}\|_2^2 \|\mathbf{N}_\alpha\|_2 = d_{\min}^{-\Delta} \|\mathbf{N}_\alpha\|_2 \tag{16}$$

For the lower bound, we exploit the fact that for symmetric matrices,

$$\|\mathbf{N}_{\alpha'}\|_2 = \max_{\|\mathbf{u}\|_2=1} |\mathbf{u}^\top \mathbf{R}\mathbf{N}_\alpha\mathbf{R}\mathbf{u}| \tag{17}$$

Setting $\mathbf{v} = \mathbf{R}\mathbf{u}/\|\mathbf{R}\mathbf{u}\|_2$ and using the fact that $\|\mathbf{R}\mathbf{u}\|_2 \geq \sigma_{\min}(\mathbf{R})\|\mathbf{u}\|_2 = d_{\max}^{-\Delta/2}$ for any unit vector $\mathbf{u}$, we obtain

$$\|\mathbf{N}_{\alpha'}\|_2 \geq \sigma_{\min}(\mathbf{R})^2 \|\mathbf{N}_\alpha\|_2 = d_{\max}^{-\Delta} \|\mathbf{N}_\alpha\|_2 \tag{18}$$

Taking absolute values and considering both positive and negative $\Delta$ yields the stated two-sided bound. The monotonicity follows when $d_{\min} \geq 1$ and $\Delta \geq 0$, since then $d_{\max}^{-\Delta} \leq d_{\min}^{-\Delta} \leq 1$. $\qquad\square$

## A.4 DERIVATIVE ANALYSIS AND DRIFT BOUNDS

We now establish the fundamental derivative relationship that governs spectral evolution and derive explicit drift bounds.

**Lemma 1** (Derivative of $\alpha$-Normalized Adjacency). *The derivative of $\mathbf{N}_\alpha$ with respect to $\alpha$ is*

$$\frac{d}{d\alpha}\mathbf{N}_\alpha = -\frac{1}{2}(\mathbf{\Gamma}\mathbf{N}_\alpha + \mathbf{N}_\alpha\mathbf{\Gamma}) \tag{19}$$

*where $\mathbf{\Gamma} = diag(\log \mathbf{d})$.*

*Proof.* Differentiate $\mathbf{N}_\alpha = \mathbf{D}_\alpha^{-1/2}\mathbf{A}\mathbf{D}_\alpha^{-1/2}$ using the product rule:

$$\frac{d}{d\alpha}\mathbf{N}_\alpha = \frac{d}{d\alpha}(\mathbf{D}_\alpha^{-1/2})\mathbf{A}\mathbf{D}_\alpha^{-1/2} + \mathbf{D}_\alpha^{-1/2}\mathbf{A}\frac{d}{d\alpha}(\mathbf{D}_\alpha^{-1/2}) \tag{20}$$

For the diagonal matrix $\mathbf{D}_\alpha = \text{diag}(d_1^\alpha, \ldots, d_n^\alpha)$, we have

$$\frac{d}{d\alpha}\mathbf{D}_\alpha^{-1/2} = \frac{d}{d\alpha}\text{diag}(d_1^{-\alpha/2}, \ldots, d_n^{-\alpha/2}) = -\frac{1}{2}\text{diag}(\log d_1 \cdot d_1^{-\alpha/2}, \ldots, \log d_n \cdot d_n^{-\alpha/2}) \tag{21}$$

This simplifies to

$$\frac{d}{d\alpha}\mathbf{D}_\alpha^{-1/2} = -\frac{1}{2}\mathbf{\Gamma}\mathbf{D}_\alpha^{-1/2} \tag{22}$$

Substituting back:

$$\frac{d}{d\alpha}\mathbf{N}_\alpha = -\frac{1}{2}\mathbf{\Gamma}\mathbf{D}_\alpha^{-1/2}\mathbf{A}\mathbf{D}_\alpha^{-1/2} - \frac{1}{2}\mathbf{D}_\alpha^{-1/2}\mathbf{A}\mathbf{D}_\alpha^{-1/2}\mathbf{\Gamma} \tag{23}$$

$$= -\frac{1}{2}(\mathbf{\Gamma}\mathbf{N}_\alpha + \mathbf{N}_\alpha\mathbf{\Gamma}) \tag{24}$$

$\qquad\square$

**Theorem 8** (Eigenvalue and Eigenvector Drift Bounds). *For any $\alpha, \alpha' \in [0, 1]$ with $\Delta = \alpha' - \alpha$, the following bounds hold:*

*(i) Matrix drift:*

$$\|\mathbf{N}_{\alpha'} - \mathbf{N}_\alpha\|_2 \leq |\Delta| \|\mathbf{\Gamma}\|_2 \sup_{t \in [\alpha, \alpha']} \|\mathbf{N}_t\|_2 \tag{25}$$

*(ii) Eigenvalue drift: For each eigenvalue $\lambda_k(\alpha)$ of $\mathbf{N}_\alpha$,*

$$|\lambda_k(\alpha') - \lambda_k(\alpha)| \leq |\Delta| \|\mathbf{\Gamma}\|_2 \sup_{t \in [\alpha, \alpha']} \|\mathbf{N}_t\|_2 \tag{26}$$

*(iii) Eigenvector drift: If the spectral gap $\gamma(t) = \lambda_1(t) - \lambda_2(t) > 0$ for all $t \in [\alpha, \alpha']$, then*

$$\sin \angle(\mathbf{y}_1(\alpha), \mathbf{y}_1(\alpha')) \leq \frac{|\Delta| \|\mathbf{\Gamma}\|_2 \sup_{t \in [\alpha, \alpha']} \|\mathbf{N}_t\|_2}{\min_{t \in [\alpha, \alpha']} \gamma(t)} \tag{27}$$

*where $\mathbf{y}_1(\alpha)$ and $\mathbf{y}_1(\alpha')$ are unit principal eigenvectors.*

*Proof.* **(i) Matrix drift:** By the fundamental theorem of calculus and Lemma 1,

$$\mathbf{N}_{\alpha'} - \mathbf{N}_\alpha = \int_\alpha^{\alpha'} \frac{d}{dt} \mathbf{N}_t \, dt = -\frac{1}{2} \int_\alpha^{\alpha'} (\mathbf{\Gamma} \mathbf{N}_t + \mathbf{N}_t \mathbf{\Gamma}) \, dt \tag{28}$$

Taking the spectral norm and using the triangle inequality:

$$\|\mathbf{N}_{\alpha'} - \mathbf{N}_\alpha\|_2 \leq \frac{1}{2} \int_\alpha^{\alpha'} \|\mathbf{\Gamma} \mathbf{N}_t + \mathbf{N}_t \mathbf{\Gamma}\|_2 \, dt \tag{29}$$

$$\leq \frac{1}{2} \int_\alpha^{\alpha'} (\|\mathbf{\Gamma} \mathbf{N}_t\|_2 + \|\mathbf{N}_t \mathbf{\Gamma}\|_2) \, dt \tag{30}$$

$$\leq \frac{1}{2} \int_\alpha^{\alpha'} 2\|\mathbf{\Gamma}\|_2 \|\mathbf{N}_t\|_2 \, dt \tag{31}$$

$$= |\Delta| \|\mathbf{\Gamma}\|_2 \sup_{t \in [\alpha, \alpha']} \|\mathbf{N}_t\|_2 \tag{32}$$

**(ii) Eigenvalue drift:** This follows directly from Weyl's inequality: for any symmetric matrix perturbation $\mathbf{E}$,

$$|\lambda_k(\mathbf{M} + \mathbf{E}) - \lambda_k(\mathbf{M})| \leq \|\mathbf{E}\|_2 \tag{33}$$

Applying this with $\mathbf{M} = \mathbf{N}_\alpha$ and $\mathbf{E} = \mathbf{N}_{\alpha'} - \mathbf{N}_\alpha$, and using part (i) gives the result.

**(iii) Eigenvector drift:** We apply the Davis-Kahan $\sin \Theta$ theorem for symmetric matrices. For simple eigenvalues separated by gap $\gamma > 0$, if $\|\mathbf{E}\|_2 < \gamma$, then

$$\sin \angle(\mathbf{y}_1, \tilde{\mathbf{y}}_1) \leq \frac{\|\mathbf{E}\|_2}{\gamma} \tag{34}$$

where $\mathbf{y}_1$ and $\tilde{\mathbf{y}}_1$ are the corresponding principal eigenvectors of $\mathbf{M}$ and $\mathbf{M} + \mathbf{E}$.

Setting $\mathbf{E} = \mathbf{N}_{\alpha'} - \mathbf{N}_\alpha$ and using the minimum gap condition along with part (i) yields:

$$\sin \angle(\mathbf{y}_1(\alpha), \mathbf{y}_1(\alpha')) \leq \frac{\|\mathbf{N}_{\alpha'} - \mathbf{N}_\alpha\|_2}{\min_{t \in [\alpha, \alpha']} \gamma(t)} \leq \frac{|\Delta| \|\mathbf{\Gamma}\|_2 \sup_{t \in [\alpha, \alpha']} \|\mathbf{N}_t\|_2}{\min_{t \in [\alpha, \alpha']} \gamma(t)} \tag{35}$$

$$\square$$

## A.5 STEP-SIZE CONTROL AND CONVERGENCE

The drift bounds immediately yield practical step-size control rules for stable annealing.

**Corollary 9** (Adaptive Step-Size Rule). *Fix $\alpha$ and a tolerance $\tau \in (0,1)$. If the spectral gap $\gamma(t) > 0$ for all $t \in [\alpha, \alpha + \Delta]$ and*

$$|\Delta| \le \tau \frac{\min_{t \in [\alpha, \alpha+\Delta]} \gamma(t)}{\|\mathbf{\Gamma}\|_2 \sup_{t \in [\alpha, \alpha+\Delta]} \|\mathbf{N}_t\|_2} \tag{36}$$

*then $\sin \angle(\mathbf{y}_1(\alpha), \mathbf{y}_1(\alpha + \Delta)) \le \tau$.*

*Furthermore, if $d_{\min} \ge 1$ and $\Delta \ge 0$, it suffices to require*

$$|\Delta| \le \tau \frac{\gamma(\alpha)}{\|\mathbf{\Gamma}\|_2 \|\mathbf{N}_\alpha\|_2} \tag{37}$$

*Proof.* The first statement follows directly from Theorem 8(iii) by setting the bound equal to $\tau$.

For the second statement, we use Theorem 7. Under the conditions $d_{\min} \ge 1$ and $\Delta \ge 0$, we have $\sup_{t \in [\alpha, \alpha+\Delta]} \|\mathbf{N}_t\|_2 \le \|\mathbf{N}_\alpha\|_2$. Additionally, for small $\Delta$, the gap $\gamma(t)$ varies continuously, so $\min_{t \in [\alpha, \alpha+\Delta]} \gamma(t) \approx \gamma(\alpha)$ for sufficiently small $\Delta$. The practical step-size rule uses these approximations. $\square$

## A.6 GLOBAL LIPSCHITZ CONTINUITY

Finally, we establish global regularity properties of the spectral family.

**Theorem 10** (Global Lipschitz Continuity of Eigenvalues). *On any compact interval $I \subset [0,1]$, each eigenvalue function $\lambda_k : I \to \mathbb{R}$ is Lipschitz continuous with constant*

$$L_I = \|\mathbf{\Gamma}\|_2 \sup_{t \in I} \|\mathbf{N}_t\|_2 \tag{38}$$

*Proof.* This follows immediately from Theorem 8(ii) by taking the supremum over all $\alpha, \alpha' \in I$:

$$|\lambda_k(\alpha') - \lambda_k(\alpha)| \le |\alpha' - \alpha| \|\mathbf{\Gamma}\|_2 \sup_{t \in I} \|\mathbf{N}_t\|_2 \tag{39}$$

The supremum $\sup_{t \in I} \|\mathbf{N}_t\|_2$ is finite since $I$ is compact and $t \mapsto \|\mathbf{N}_t\|_2$ is continuous. $\square$

## A.7 ITERATION COMPLEXITY OF WARM-STARTING

The diagonal predictor reduces computational cost by providing accurate initial guesses for successive eigenproblems. We now establish theoretical bounds on the iteration count reduction.

**Lipschitz continuity of $N_\alpha$.** The operator $N_\alpha = D^{-\alpha/2} A D^{-\alpha/2}$ depends smoothly on $\alpha$ through the diagonal scaling. Since $D^{-\alpha/2} = \text{diag}(d_i^{-\alpha/2})$ and $\frac{\partial}{\partial \alpha} d^{-\alpha/2} = -\frac{1}{2} d^{-\alpha/2} \log d$, we have

$$D^{-(\alpha+\Delta\alpha)/2} = D^{-\alpha/2} \left( I - \frac{\Delta\alpha}{2} \log D + O(\Delta\alpha^2) \right),$$

where $\log D = \text{diag}(\log d_i)$. This yields the Lipschitz bound

$$\|N_{\alpha+\Delta\alpha} - N_\alpha\| \le L|\Delta\alpha|, \quad \text{where} \quad L = O\left( \|A\| \cdot \max_i d_i^{-1} |\log d_i| \right). \tag{40}$$

The constant $L$ depends only on graph structure, not problem dimension.

**Warm-start quality via Davis–Kahan.** Let $u(\alpha)$ denote the top eigenvector of $N_\alpha$ and $\gamma(\alpha) = \lambda_1(\alpha) - \lambda_2(\alpha)$ the spectral gap. Applying the Davis–Kahan $\sin \Theta$ theorem (Davis & Kahan, 1970) to equation 40 yields

$$\theta_0 \equiv \angle\big(u(\alpha + \Delta\alpha), u(\alpha)\big) \lesssim \frac{L|\Delta\alpha|}{\gamma(\alpha)}. \tag{41}$$

Thus the eigenvector at parameter $\alpha$ provides a warm start whose initial angular error is proportional to $|\Delta\alpha|/\gamma(\alpha)$.

**Iteration count bounds.** For eigenvalue methods based on power iteration or Krylov subspaces, convergence from initial angle $\theta_0$ follows

$$\theta_m \leq q_{\text{spec}}^m \theta_0, \quad \text{where} \quad q_{\text{spec}} = \sup_{\alpha\in[0,1]} \frac{|\lambda_2(\alpha)|}{|\lambda_1(\alpha)|} < 1$$

is the worst-case contraction factor. To reach accuracy $\varepsilon$, we require

$$m \leq \frac{\log(L|\Delta\alpha|/(\gamma(\alpha)\varepsilon))}{\log(1/q_{\text{spec}})}. \tag{42}$$

Hence the iteration count grows only *logarithmically* with $|\Delta\alpha|/\gamma(\alpha)$ and remains $O(1)$ when the schedule satisfies $|\Delta\alpha| \ll \gamma(\alpha)$. Averaging along the continuation path yields

$$\bar{I} = O\left(1 + \log\frac{|\Delta\alpha|}{\gamma(\alpha)}\right). \tag{43}$$

**Practical implementation with thick-restart Lanczos.** The bounds above assume ideal unrestarted iterations. In practice, `cupyx.scipy.sparse.linalg.eigsh` uses thick-restart Lanczos with fixed Krylov dimension $n_{\text{cv}}$ (default $n_{\text{cv}} = 33$ for $k = 1$). This imposes a structural floor: even when $\theta_0$ is very small, the method must complete at least $n_{\text{cv}}$ iterations before checking convergence. When warm starts are sufficiently accurate, convergence occurs within the first restart cycle, creating an empirical plateau at $n_{\text{cv}}$ iterations. This explains the consistent $\approx 33$ iteration count observed across problem sizes.

# B  APPENDIX: ALGORITHM

## B.1  IMPLEMENTATION DETAIL

The eigensolver is built on GPU-accelerated sparse linear algebra routines, providing efficient support for large-scale eigenvalue problems. At its core, the method relies on an iterative Krylov subspace procedure for real symmetric sparse matrices, where successive Ritz pairs are refined until convergence.

To accelerate convergence, we employ a warm-start strategy in which the leading eigenvector obtained at the previous $\alpha$ value is reused as the initialization for the next step. This avoids restarting from random vectors, thereby reducing the number of iterations required and significantly improving runtime. The benefit is particularly pronounced in our framework, since consecutive $\alpha$ values define closely related eigenproblems, allowing the solver to exploit temporal continuity.

To enhance the quality of the rounded solutions obtained from the spectral relaxation, we integrate a refinement stage based on GPU-accelerated batch local search. This procedure simultaneously optimizes all candidate spin configurations produced along the $\alpha$ sweep, enabling high throughput and consistency in the refinement phase. The refinement is formulated as an iterative spin-flip process: for each candidate $s \in {-1, +1}^n$, the local field acting on spin $i$ is computed as $l_i = \sum_j A_{ij} s_j$, and a flip is applied whenever $\Delta E = 2 s_i l_i < 0$. By exploiting warp-level parallelism, thousands of spins across multiple solutions can be updated in parallel, reducing refinement time by orders of magnitude compared to sequential CPU implementations. To prevent premature stagnation, we incorporate randomized subset selection and reshuffled multi-round sweeps, which diversify flip sequences while ensuring monotone descent in energy. This refinement step consistently improves solution quality at scale.

The local refinement is executed through repeated kernel launches that update batches of spins across multiple candidate solutions simultaneously. At each iteration, a random subset of nodes is selected, and the CUDA kernel computes their local fields in parallel using warp-level reductions. A flip is applied if $\Delta E < 0$, ensuring monotone descent. This design allows thousands of spin updates to be processed concurrently, maximizing GPU occupancy and amortizing memory transfers. The procedure iterates until no further improvements are detected or a maximum number of iterations is reached. A final sweep over all nodes guarantees convergence. Algorithm 2 summarizes this process, showing how kernel calls are orchestrated within the refinement loop to preserve the logic of sequential spin-flip heuristics while fully exploiting GPU parallelism.

---

**Algorithm 1** Spectral Annealing for Ising Optimization

---

**Require:** Signed adjacency matrix $\mathbf{A} \in \mathbb{R}^{n \times n}$
**Require:** Annealing schedule $\{\alpha_k\}_{k=0}^K$, tolerance $\tau \in (0, 1)$
**Ensure:** Best configuration $\mathbf{s}^*$ and energy $E^*$
1: Initialize: $E^* \leftarrow +\infty$, $\mathbf{s}^* \leftarrow$ random, $\mathbf{y}_0 \leftarrow$ random unit vector
2: Compute $\mathbf{\Gamma} = \mathrm{diag}(\log \mathbf{d})$, degree bounds $d_{\min}, d_{\max}$
3: **for** $k = 0, 1, \ldots, K$ **do**
4:      **// Spectral shift for positive definiteness**
5:      $\rho_k \leftarrow (1 + 0.1) \max_i d_i^{1-\alpha_k}$
6:      **// Diagonal predictor warm-start (skip for $k = 0$)**
7:      **if** $k > 0$ **then**
8:          $\Delta \leftarrow \alpha_k - \alpha_{k-1}$
9:          $\mathbf{y}_{\mathrm{pred}} \leftarrow \frac{\mathbf{D}^{\Delta/2}\mathbf{y}_{k-1}}{\|\mathbf{D}^{\Delta/2}\mathbf{y}_{k-1}\|_2}$
10:      **else**
11:          $\mathbf{y}_{\mathrm{pred}} \leftarrow \mathbf{y}_0$
12:      **end if**
13:      **// Solve smallest eigenpair of $\mathbf{S}_{\rho_k}(\alpha_k) = \rho_k \mathbf{I} - \mathbf{N}_{\alpha_k}$**
14:      $(\nu_k, \mathbf{y}_k) \leftarrow \textsc{Lanczos}(\mathbf{S}_{\rho_k}(\alpha_k), \mathbf{y}_{\mathrm{pred}})$
15:      $\lambda_k \leftarrow \rho_k - \nu_k$ **// Recover original eigenvalue**
16:      **// Round to Ising spins**
17:      $\hat{\mathbf{s}}_k \leftarrow \mathrm{sign}(\mathbf{D}_{\alpha_k}^{-1/2}\mathbf{y}_k)$
18:      **// Local 1-flip improvement**
19:      $\hat{\mathbf{s}}_k \leftarrow \textsc{LocalSearch}(\hat{\mathbf{s}}_k, \mathbf{A})$
20:      **// Track best solution**
21:      $E_k \leftarrow -\frac{1}{2}\hat{\mathbf{s}}_k^\top \mathbf{A} \hat{\mathbf{s}}_k$
22:      **if** $E_k < E^*$ **then**
23:          $E^* \leftarrow E_k$, $\mathbf{s}^* \leftarrow \hat{\mathbf{s}}_k$
24:      **end if**
25:      **// Adaptive step-size for next iteration**
26:      **if** $k < K$ **then**
27:          Compute spectral gap $\gamma_k = \lambda_k - \lambda_2(\mathbf{N}_{\alpha_k})$
28:          $\Delta_{\max} \leftarrow \tau \frac{\gamma_k}{\|\mathbf{\Gamma}\|_2 \|\mathbf{N}_{\alpha_k}\|_2}$
29:          $\alpha_{k+1} \leftarrow \min(\alpha_k + \Delta_{\max}, 1)$
30:      **end if**
31: **end for**
32: **return** $\mathbf{s}^*, E^*$

---

The CUDA kernel implementation exploits the hierarchical GPU architecture. Candidate spin configurations are stacked into a flat $1D$ tensor and distributed across a grid of thread blocks, with each block processing a subset of nodes and solutions. Inside each block, warps are assigned to individual nodes, and the 32 threads collaboratively traverse adjacency lists in the CSR format. Partial contributions to the local field are computed in parallel and aggregated through warp-level shuffle reductions. If the resulting energy change is negative, the flip is executed by Thread 0, updating the global spin vector in place.

## C    Appendix: Experiments

### C.1    Setup

**Datasets.** We evaluate on both established benchmarks and large-scale synthetic instances designed to assess scalability:

---

**Algorithm 2** GPU Batch Local Search

---

**Require:** Sparse matrix $\mathbf{A}$ in CSR format ($indptr, indices, data$), spin solutions $\{s^{(1)}, \ldots, s^{(k)}\}$, maximum iterations $T$, subset fraction $\phi$

**Ensure:** Refined spin configurations $\{s_{\text{ref}}^{(1)}, \ldots, s_{\text{ref}}^{(k)}\}$

 1: Initialize: stack spins into flat array $\mathbf{s} \in \{-1, +1\}^{nk}$, set $improvement\_counts \leftarrow 0$, $active\_mask \leftarrow$ **True**
 2: Define CUDA kernel UPDATESPINS for local field computation and flips
 3: **for** $t = 1$ to $T$ **do**
 4:    $num\_selected \leftarrow \lfloor \phi \cdot n \rfloor$
 5:    $selected\_nodes \leftarrow$ random permutation of $\{0, 1, \ldots, n-1\}$ truncated to $num\_selected$
 6:    Reset $improvement\_counts \leftarrow 0$
 7:    **// Parallel kernel launch over all solutions and selected nodes**
 8:    UPDATESPINS($selected\_nodes, \mathbf{A}, \mathbf{s}, improvement\_counts$)
 9:    Update $active\_mask \leftarrow active\_mask \wedge (improvement\_counts > 0)$
10:    **if** all solutions inactive **then**
11:       **break**
12:    **end if**
13: **end for**
14: **// Final multi-round sweep to ensure convergence**
15: **for** $r = 1$ to $num\_rounds$ **do**
16:    $nodes \leftarrow$ random permutation of all $n$ nodes
17:    **for** $sub = 1$ to $max\_sub\_iters$ **do**
18:       UPDATESPINS($nodes, \mathbf{A}, \mathbf{s}, improvement\_counts$)
19:       **if** no improvements in any solution **then**
20:          **break**
21:       **end if**
22:    **end for**
23: **end for**
24: Reshape $\mathbf{s}$ into $\{s_{\text{ref}}^{(1)}, \ldots, s_{\text{ref}}^{(k)}\}$
25: **return** Refined spin configurations

---

*Gset benchmark* (Ye, 2003). We use 66 instances from the widely-adopted Gset collection (excluding 5 disconnected graphs), ranging from 800 to 40,000 nodes. These instances span diverse structural properties, including both dense and sparse random graphs, providing a standard comparison baseline for Ising optimization methods.

*Large-scale synthetic graphs*. To assess scalability beyond existing benchmarks, we generate two complementary graph families: Barabási-Albert (BA), scale-free graphs with preferential attachment ($m = 20$), producing heavy-tailed degree distributions typical in social networks and citation graphs. Erdős-Rényi (ER), homogeneous random graphs with edge probabilities matched to achieve a similar density as corresponding BA graphs, representing more uniform connectivity patterns. Sizes range from $n = 64$ to $n = 8,388,608$ nodes. Edge weights are drawn uniformly from $[-100, 100]$ to ensure non-trivial optimization landscapes with mixed ferromagnetic and antiferromagnetic interactions. For each configuration, we generate ten independent instances to account for stochastic variability.

*Real-world signed networks*. We further evaluate on three large-scale social networks with explicit trust/distrust or friend/foe relations, widely studied in signed network analysis (Leskovec et al., 2010). Epinions is a trust network where users label others as trustworthy or not. Slashdot captures friend/foe relationships from the Slashdot technology community. Wikipedia elections (WikiElec) record votes of support or opposition in administrator elections. While all three encode positive and negative social ties, they differ in context: consumer trust (Epinions), online community endorsements (Slashdot), and collaborative governance (Wikipedia). To cast them as Ising problems, we map each node to a spin $s_i \in \{-1, +1\}$ and assign couplings $J_{ij} = +1$ for positive edges (trust/friend/support) and $J_{ij} = -1$ for negative edges (distrust/foe/oppose), with missing edges treated as $J_{ij} = 0$. Minimizing the resulting Ising Hamiltonian thus seeks spin assignments consistent with observed social agreements and disagreements.

**Benchmarked Solvers.** To evaluate the performance of our SpecAnn approach, we compare it against a diverse set of solvers spanning exact optimization, metaheuristics, constructive heuristics, and spectral relaxations. As an exact reference, we include the commercial solver Gurobi (Gurobi Optimization, LLC, 2023), limited to 2 hours per instance

and configured with `MIPFocus=1` to emphasize high-quality feasible solutions over formal optimality proofs. We also evaluate metaheuristics such as Simulated Annealing (SA) (Kirkpatrick et al., 1983), implemented via D-Wave's `neal` library D-Wave Systems Inc. (2022), with 100 samples and default values for all other parameters; Simulated Bifurcation (SB) (Goto et al., 2019), which exploits quantum-inspired oscillator dynamics and GPU parallelism (heated discrete SB with 1024 agents); SPONGE (Cucuringu et al., 2019) implemented using normalized symmetric version with default parameters; Tabu Search (TS) (Glover, 1989), incorporating short-term memory through the MST2 algorithm in D-Wave's `tabu` D-Wave Systems Inc. (2019); and Parallel Tempering (PT) (Earl & Deem, 2005), implemented via omnisolver-pt library (Quantum AI Lab, 2022). As a constructive heuristic, we adopt a simple Greedy descent, repeated over 1000 random restarts to mitigate initialization bias. Regarding the recent Quasi-Quantum Annealing (QQA) Ichikawa & Arai (2024), we perform the experiments with a batch size of 100 parallel solutions and a small temperature (0.001) for noise-based exploration. The QQA annealing schedule's background field range was -3.0 to 0.1, with an increased penalty curve rate of 4.0 and a diversity parameter of 0.2 to encourage solution diversity, and 3000 iterations. Finally, we include two classical spectral relaxations, Adjacency Eigenvalue (SpecAdj) where $S = I - J$ and the Normalized Signed Laplacian (SpecSignedLap) where $S = D - J$, both of which approximate the ground state by rounding eigenvector solutions. Spectral methods used `eigsh` from `cupyx sparse` library with eigenvalue tolerance $10^{-6}$ and maximum 1000 Arnoldi iterations. Local search was limited to 1000 iterations for consistency across methods.

**Hyperparameter Tuning.** To ensure fair comparison, we systematically tuned hyperparameters for all baseline methods using the same BA instances (sizes $n = 2^6$ to $2^{20}$) and a 2-hour time limit per instance. Our tuning protocol prioritized configurations that balance solution quality and computational efficiency, as methods with excessive runtime would not be practically competitive even if they achieved marginally better solutions.

*Simulated Annealing (SA).* SA was implemented using the `dwave-neal` solver, which requires two critical parameters: the number of independent samples (parallel trajectories) and the number of sweeps per trajectory (annealing depth) (Kirkpatrick et al., 1983; Geman & Geman, 1984; Hajek, 1988). Following Tasseff et al. (2024), we fixed the number of samples at 100 to maintain computational feasibility while exploring sweep counts of 100, 1000, and 10000.

Solution quality scaled monotonically with sweep count, but with diminishing returns. The 100-sweep configuration converged quickly but produced energy gaps of 3–4% relative to best-known solutions. The 10000-sweep configuration achieved the best quality but required approximately $10\times$ longer runtime than 1000 sweeps. The 1000-sweep configuration consistently achieved energy gaps below 1% while maintaining reasonable runtime, making it the optimal choice for balancing quality and efficiency. All SA experiments used this configuration.

*Simulated Bifurcation (SB).* SB offers two dynamical models (discrete and ballistic) and an optional Nosé–Hoover thermostat for thermal fluctuations (Goto et al., 2019; Kanao & Goto, 2022). We evaluated all four combinations on BA instances. The thermostat proved essential: thermal fluctuations reduced energy gaps by up to 12% on the $n = 16{,}384$ instance compared to deterministic dynamics, consistent with findings by Kanao & Goto (2022). Between the two models, the discrete variant exhibited 15–20% faster runtime on large graphs ($n \geq 32{,}768$) with negligible quality difference. Consequently, all SB experiments used the heated discrete model on GPU.

*Quasi-Quantum Annealing (QQA).* QQA was evaluated on BA instances up to $n = 65{,}536$ using two tuning strategies: exploratory grid search and configurations from Ichikawa & Arai (2024). For exploratory tuning, we varied the learning rate $\eta \in \{0.8, 1.0, 1.2\}$, diversity parameter $d \in \{0.15, 0.2, 0.25, 0.3, 0.4\}$, and epoch count $E \in \{2500, 3000, 4000, 6000\}$. The diversity parameter exhibited the strongest influence on solution quality, controlling exploration across parallel runs. While $d = 0.15$ achieved marginally better quality than $d = 0.2$ (energy gap difference $< 0.5\%$), it incurred approximately $2\times$ runtime overhead on large instances ($n \geq 32{,}768$). We selected $d = 0.2$, default learning rate, and $E = 3000$ epochs to ensure convergence without excessive runtime.

For paper-based configurations, we followed Ichikawa & Arai (2024) and varied the background field range $[\gamma_{\min}, \gamma_{\max}]$ with $\gamma_{\max} = 0.1$ fixed and $\gamma_{\min} \in \{-2, -3, -5, -10, -20\}$. While $\gamma_{\min} = -20$ yielded the best solution quality, runtime increased progressively with $|\gamma_{\min}|$, reaching $2.6\times$ slower than $\gamma_{\min} = -3$ for $n = 65{,}536$. The configuration $\gamma_{\min} = -3$ provided the best quality-runtime tradeoff and was used for all subsequent experiments. All QQA experiments were performed on GPU using the implementation from Ichikawa & Arai (2024).

**Computational Environment.** All experiments were conducted on Intel Xeon Platinum 8276L processors (2.20GHz) for CPU-based experiments, while GPU experiments employed NVIDIA A100-PCIE-40GB accelerators.

Table 4: The time-to-target (in seconds, with reference to SpecAnn) across Gset instances.

| | QQA | SA | GUROBI | SB | SpecAnn | | QQA | SA | GUROBI | SB | SpecAnn |
|---|---|---|---|---|---|---|---|---|---|---|---|
| G1 | 2.39 | **0.53** | 8.07 | 1.20 | 2.27 | G34 | 12.26 | 0.43 | **0.14** | 1.01 | 1.68 |
| G2 | 1.39 | **0.45** | 14.62 | 1.03 | 3.32 | G35 | 20.48 | **0.60** | 491.80 | 1.12 | 3.33 |
| G3 | 1.39 | **0.47** | 12.81 | 0.84 | 2.07 | G36 | 17.89 | **0.58** | 263.64 | 1.16 | 3.41 |
| G4 | 1.39 | **0.44** | 7.06 | 0.83 | 2.07 | G37 | 21.76 | **0.64** | 283.55 | 1.15 | 3.16 |
| G5 | 1.39 | **0.52** | 15.39 | 0.85 | 2.06 | G38 | 20.47 | **0.63** | 577.92 | 1.16 | 3.02 |
| G6 | 1.39 | **0.45** | 8.86 | 0.73 | 1.75 | G39 | 1.28 | **0.62** | 115.79 | 1.11 | 2.11 |
| G7 | 1.39 | **0.48** | 7.89 | 0.82 | 2.01 | G40 | 1.28 | **0.71** | – | 1.04 | 2.45 |
| G8 | 1.39 | **0.48** | 11.13 | 0.79 | 1.76 | G41 | 1.28 | **0.64** | 104.46 | 1.05 | 2.25 |
| G9 | 1.39 | **0.47** | 9.44 | 0.82 | 1.91 | G42 | 1.28 | **0.57** | – | 1.16 | 2.19 |
| G10 | 1.39 | **0.48** | 13.79 | 0.85 | 3.11 | G43 | 1.18 | **0.34** | – | 0.98 | 4.23 |
| G11 | 11.35 | 0.19 | **0.05** | 0.99 | 1.70 | G44 | 1.18 | **0.40** | – | 0.96 | 2.53 |
| G12 | 10.33 | 0.18 | **0.05** | 0.64 | 1.69 | G45 | 1.18 | **0.35** | 4.30 | 0.98 | 2.13 |
| G13 | 10.32 | 0.15 | **0.10** | 0.73 | 1.72 | G46 | 1.18 | **0.31** | – | 0.96 | 3.14 |
| G14 | 15.62 | **0.22** | 336.70 | 1.00 | 2.95 | G47 | 1.18 | **0.34** | – | 0.80 | 3.14 |
| G15 | 20.09 | **0.24** | 2580.28 | 0.95 | 2.74 | G48 | 1.21 | **0.54** | 0.91 | 1.46 | 1.68 |
| G16 | 5.58 | 0.21 | **0.12** | 0.80 | 2.33 | G49 | 1.21 | **0.55** | 1.09 | 1.50 | 1.73 |
| G17 | 15.62 | **0.25** | 33.19 | 0.75 | 3.14 | G50 | 1.21 | – | **0.56** | – | 1.73 |
| G18 | 1.12 | **0.27** | 2750.94 | 0.86 | 2.66 | G51 | 23.83 | **0.34** | – | 0.81 | 3.22 |
| G19 | 1.13 | **0.27** | – | 0.97 | 2.32 | G52 | 19.30 | **0.29** | 57.16 | 1.01 | 3.09 |
| G20 | 1.12 | **0.25** | – | 0.74 | 2.20 | G53 | 17.04 | **0.29** | 231.30 | 0.81 | 3.27 |
| G21 | 6.70 | **0.27** | – | 0.77 | 2.35 | G54 | 20.45 | **0.35** | – | 1.01 | 3.16 |
| G22 | 1.54 | **0.76** | – | 1.15 | 2.12 | G57 | 18.28 | **0.93** | 0.93 | 2.77 | 1.73 |
| G23 | 1.54 | **0.81** | – | 1.11 | 2.55 | G58 | 28.10 | **2.08** | 296.20 | 2.83 | 3.18 |
| G24 | 1.54 | **0.72** | – | 1.14 | 3.01 | G59 | 2.35 | **1.92** | 993.79 | 2.68 | 2.91 |
| G25 | 1.54 | **0.83** | – | 1.11 | 3.60 | G62 | 24.25 | **1.35** | 3.04 | 5.19 | 1.71 |
| G26 | 1.54 | **0.81** | – | 1.09 | 3.57 | G63 | 44.36 | **2.78** | 2209.43 | 10.76 | 3.24 |
| G27 | 1.54 | **0.80** | – | 1.18 | 2.30 | G64 | 9.51 | 2.87 | – | 6.50 | **2.66** |
| G28 | 1.54 | **0.83** | – | 1.05 | 3.03 | G65 | 29.36 | **1.48** | – | 6.03 | 1.69 |
| G29 | 1.54 | **0.66** | 11.27 | 1.05 | 2.70 | G66 | 28.97 | 2.08 | – | 7.80 | **1.70** |
| G30 | 1.54 | **0.67** | – | 1.12 | 2.08 | G67 | 31.34 | 2.24 | – | 9.33 | **1.74** |
| G31 | 1.54 | **0.81** | – | 1.09 | 2.75 | G72 | 31.31 | 2.07 | – | 9.54 | **1.68** |
| G32 | 12.20 | **0.38** | 10.59 | 1.07 | 1.68 | G77 | 40.89 | 3.15 | – | 17.64 | **1.67** |
| G33 | 13.33 | 0.43 | **0.32** | 1.09 | 1.88 | G81 | 60.49 | 3.59 | – | 34.23 | **1.66** |

Table 5: The time-to-target (in seconds, with reference to SpecAnn) across BA instances.

| Size ($n$) | #edges | QQA | SA | GUROBI | SB | SpecAnn |
|---:|---:|---:|---:|---:|---:|---:|
| 64 | 880 | – | **0.03** | 0.04 | 0.60 | 1.36 |
| 128 | 2160 | 1.32 | **0.06** | 0.19 | 0.64 | 1.36 |
| 256 | 4720 | – | **0.12** | – | 0.66 | 1.40 |
| 512 | 9840 | – | **0.23** | – | 0.69 | 1.66 |
| 1024 | 20080 | – | **0.47** | – | 1.07 | 1.44 |
| 2048 | 40560 | – | 1.30 | – | **0.84** | 1.83 |
| 4096 | 81520 | – | 2.58 | – | 2.50 | **1.85** |
| 8192 | 163440 | – | 5.16 | – | 7.97 | **2.02** |
| 16384 | 327280 | – | 29.54 | – | 29.54 | **2.33** |
| 32768 | 654960 | – | 67.23 | – | – | **2.32** |
| 65536 | 1310320 | – | 181.53 | – | – | **3.06** |
| 131072 | 2621040 | – | 450.84 | – | – | **3.83** |
| 262144 | 5242480 | – | 1129.80 | – | – | **5.80** |
| 524288 | 10485360 | – | 2439.86 | – | – | **12.72** |
| 1048576 | 20971120 | – | – | – | – | **27.28** |

## C.2 Time-to-target analysis

Regarding the interpretability of heterogeneous runtime and solution-quality metrics, we carried out a time-to-target analysis using SpecAnn's solution quality as the target value. For each competing method, we measured how long it took to reach or surpass SpecAnn's quality on each instance. Methods that did not reach the target within one hour were marked as unsuccessful. Because Greedy, TS, and PT performed poorly in earlier experiments, they were omitted from this part of the study.

For the Gset benchmark (Table 4), the evaluation procedure was tailored to each method's computational behavior. For Gurobi, we tracked progress by recording the time whenever a new incumbent solution was found. QQA was sampled every 500 epochs. SA was run repeatedly with increasing sample sizes, beginning with 10 samples and increasing by 10 until the target was met. Similarly, SB was evaluated starting with 100 agents and increasing by 100 in each run. All methods used the same hyperparameters as in earlier experiments to maintain consistency.

The results exhibit clear trends across problem sizes. On small and medium instances (up to roughly 10,000 spins), SA is extremely fast and often reaches the target in well under a second. For larger instances (beyond 20,000 spins), SpecAnn offers the best balance between solution quality and computation time, consistent with our earlier findings on its scalability. Gurobi performs well on a subset of instances whose structure suits branch-and-bound search, but often fails to converge within one hour on larger or structurally harder cases. QQA shows mixed behavior: it excels on some instance families but struggles on others, reflecting the sensitivity of gradient-based methods to problem structure.

The BA instances further illustrate scalability differences. To keep the experiments manageable, we selected one representative graph per size. As shown in Table 5, SA's time-to-target grows rapidly with graph size, taking more than 2,400 seconds on the half-million-node instance, compared with SpecAnn's 13 seconds. Gurobi and SB remain viable only on small graphs (fewer than 10,000 nodes), and both consistently fail to meet the target on larger ones. QQA was unable to match SpecAnn's solution quality under the same hyperparameter settings used previously. This gap is not unexpected: unlike the Gset graphs, our synthetic BA graphs are relatively dense (average degree $\approx 20$) and represent general Ising instances with edge weights drawn uniformly from $[-100, 100]$, rather than the $\pm 1$ max-cut structure of Gset. These properties make the loss landscape substantially more rugged and amplify gradient-based instability in QQA. Although additional hyperparameter tuning might improve QQA's performance on these harder instances, pursuing such optimization is beyond the scope of this work and would give QQA an unfair advantage relative to the other methods.

### C.2.1 Scalability results on Erdős-Rényi

The results of Erdős-Rényi (ER) dataset are shown in Fig. 4.

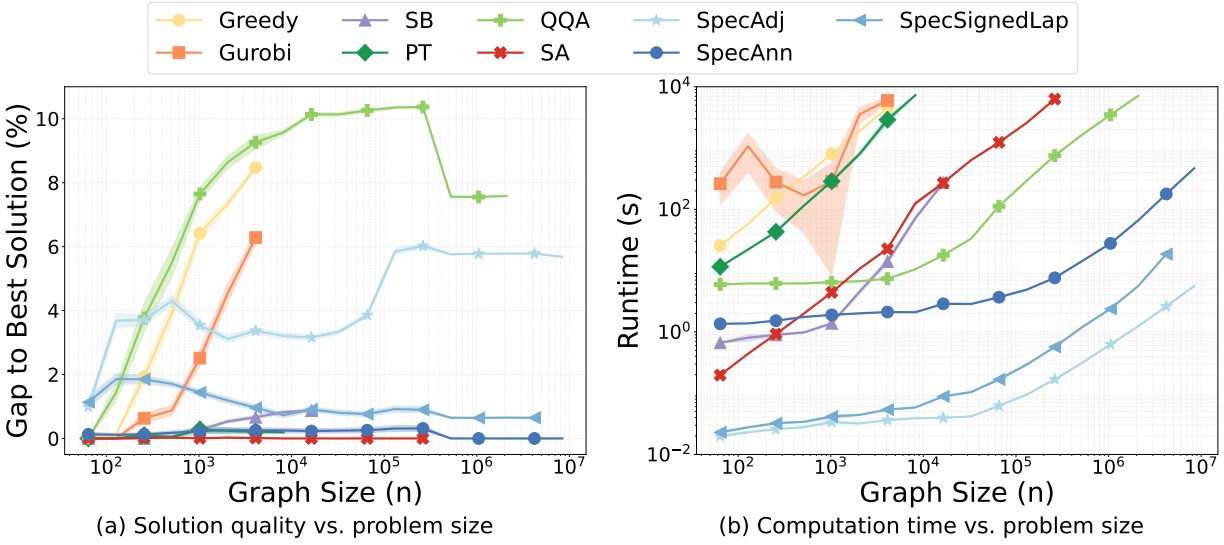

(a) Solution quality vs. problem size

(b) Computation time vs. problem size

Figure 4: Scalability results on Erdős-Rényi (ER) random graphs. The homogeneous structure of ER graphs benefits all spectral methods, with SpecAnn providing improved quality over single-shot approaches.

### C.3 GPU-ACCELERATION FOR EIGENSOLVER AND BATCHED LOCAL SEARCH

To assess implementation efficiency, we conducted an ablation study comparing the default SpecAnn against three variants: SpecAnn[-FastEig] disabling warm-starting, SpecAnn[-FastL] replacing batched with sequential local search, and SpecAnn[-FastL,-FastEig] disabling both optimizations. Figure 5 reveals a clear optimization hierarchy. FastEig warm-starting provides moderate but consistent gains, with SpecAnn[-FastEig] showing 1.2–2× penalty across all scales by exploiting the mathematical structure of consecutive $\alpha$ eigenproblems. FastL batched local search emerges as the critical component: SpecAnn[-FastL] suffers 2–4× degradation that worsens dramatically with scale, reaching 10× penalty at $n = 10^5$. For $n \geq 10^6$, performance differences between variants diminish as the GPU memory bottleneck dominates computational costs, equalizing runtimes across all implementations. This batching proves essential for GPU parallelism, amortizing kernel launches and maximizing parallel processing. The worst-case SpecAnn[-FastL,-FastEig] combines both penalties, showing 3–15× runtime degradation with near-exponential growth. The 10–15× performance difference between variants demonstrates that implementation quality fundamentally determines algorithmic competitiveness—representing the difference between SpecAnn's favorable time-quality position versus complete unviability. This validates that the method's superior performance stems from integrating mathematical insight with high-performance computing optimization.

### C.3.1 SPONGE RESULTS ON BARABÁSI-ALBERT

Because SPONGE performs far worse in terms of solution quality, we report its results separately from the main comparisons. As shown in Table 6, its energy gaps remain above 14% even on small problems and exceed 50% on the largest ones, whereas the other methods generally stay below 10%. For instance, at $n = 262,144$, SPONGE's gap to the best recorded solution is $53.82 \pm 12.51\%$, while SpecAnn reaches a near-optimal $0.35 \pm 0.11\%$. Although SPONGE is extremely fast - finishing in about half a minute even on one-million-node graphs - this speed comes at the cost of accuracy. For this reason, we leave SPONGE out of the detailed performance comparisons and focus on methods that offer a more meaningful balance between runtime and solution quality.

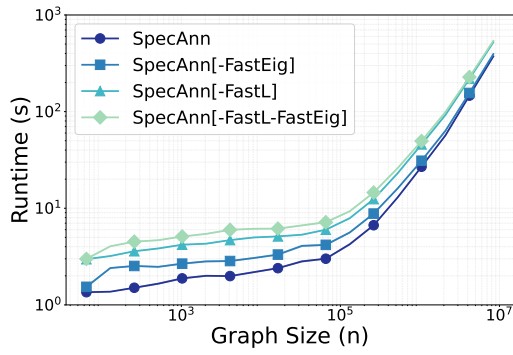

Figure 5: Computation time (s) on Barabási-Albert (BA) instances.

Table 6: Average gaps to best solutions (%) and running time of SPONGE on Barabási-Albert (BA) random graphs

| $n$ | $2^6$ | $2^7$ | $2^8$ | $2^9$ | $2^{10}$ | $2^{11}$ | $2^{12}$ | $2^{13}$ | $2^{14}$ | $2^{15}$ | $2^{16}$ | $2^{17}$ | $2^{18}$ | $2^{19}$ | $2^{20}$ | $2^{21}$ | $2^{22}$ | $2^{23}$ |
|---|---|---|---|---|---|---|---|---|---|---|---|---|---|---|---|---|---|---|
| **Gap (%)** | 14.2 | 19.2 | 16.9 | 20.4 | 22.4 | 22.2 | 22.6 | 22.4 | 22.8 | 22.9 | 30.7 | 46.2 | 53.8 | 28.3 | 20.9 | 22.8 | 29.4 | 42.2 |
| **Time (s)** | 0.01 | 0.02 | 0.02 | 0.03 | 0.04 | 0.06 | 0.10 | 0.17 | 0.33 | 0.75 | 1.5 | 3.4 | 7.4 | 14 | 33 | 79 | 170 | 561 |

## C.4 LOCAL SEARCH IMPACT

To quantify the role of the local search component, we perform an empirical study comparing SpecAnn with and without refinement. Note that in this ablation we completely disable local search, which is different from the SpecAnn`[-FastL]` variant, where the local search is applied to one solution at a time.

We consider a family of Barabási–Albert graphs with sizes ranging from $n = 2^6$ to $n = 2^{20}$, since the effect of local search may be particularly relevant at large scale. All variants use exactly the same hyperparameters and $\alpha$-schedule to ensure a fair comparison. In the SpecAnn`[-LS]` configuration, the spin configuration is obtained directly from the sign of the final eigenvector at each $\alpha$; the best energy across the continuation sweep is then reported. To further disentangle the contribution of the spectral component and the refinement stage, we also include a baseline `LS`, in which 128 random spin configurations (simulating the number of $\alpha$ evaluations in SpecAnn) are generated and post-processed using the *same* GPU-based local search as the full solver. This comparison isolates the value of the local improvement procedure independently of the spectral relaxation.

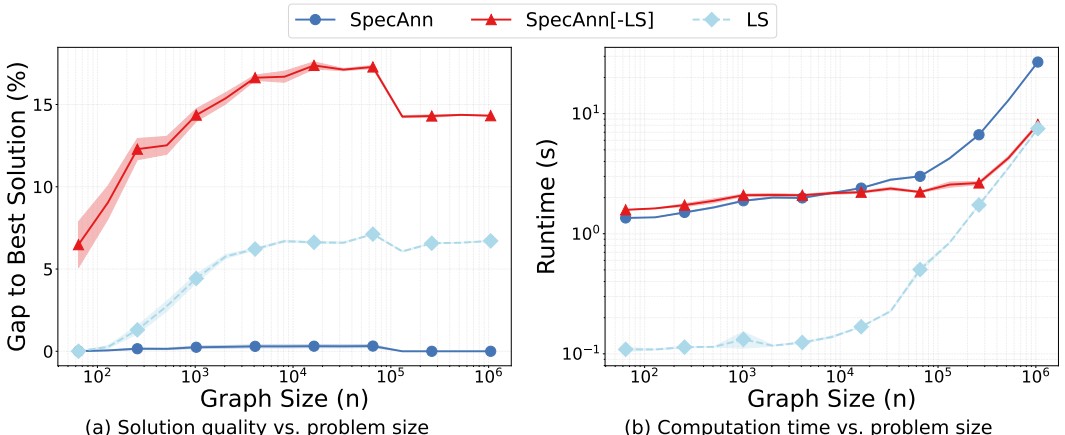

Figure 6: Impact of local search on BA graphs

Figure 6 shows that local search has a substantial effect on solution quality across all problem sizes. Disabling refinement (SpecAnn`[-LS]`) leads to significantly worse energies, with gaps between $5\%$ and $17\%$, whereas the full SpecAnn pipeline consistently achieves lower percent gaps even at $n = 10^6$. The `LS` baseline performs better than SpecAnn`[-LS]`, confirming that refinement is essential, but still remains far from the performance of SpecAnn, demonstrating that the spectral relaxation is crucial for providing good starting configurations, especially at large scale.

A noticeable drop in the energy gap for all methods around $n \approx 10^5$ arises from the scalability limitations of the baselines: some configurations cannot be executed reliably beyond that size, or they fail to improve upon the solution already produced by SpecAnn. Hence, only SpecAnn provides meaningful results in the upper end of the problem range.

On the runtime plot, SpecAnn and SpecAnn`[-LS]` have nearly identical costs, since the local search accounts for only a small fraction of the total pipeline time. The `LS`-only baseline is the fastest for small and medium problem sizes, as expected, but its refinement stage requires substantially more iterations due to the poor random initializations. As a result, its runtime grows more quickly and becomes less competitive as $n$ increases. This highlights the importance of providing high-quality initial configurations for enabling fast convergence of the post-processing step.

Overall, these results confirm that (i) the local refinement stage is critical for attaining high-quality solutions, and (ii) SpecAnn achieves a favourable accuracy–runtime tradeoff, preserving solution quality at scale while adding only a negligible computational overhead.

## C.5 EMPIRICAL UNIVERSAL BOUND ANALYSIS

To quantify the theoretical bound advantage of the $\alpha$-parameterization, we compared the tightness of the derived energy bounds (Eq. 5) against the bound provided by SpecAdj ($\alpha = 0$) and SpecSignedLap ($\alpha = 1$), as well as the ground truth computed via Gurobi (small set of BA graphs with $n \in [20, 60]$). Figure 7 displays the normalized energy ratios for varying graph sizes.

The SpecAdj bound provides the loosest relaxation, deviating significantly from the true optimum (ratios $> 1.4$). The SpecSignedLap bound offers a noticeable improvement, but the optimized SpecAnn bound ($\alpha_{opt}$) consistently yields the tightest relaxation. Notably, the true optimum (1.0) lies between the SpecAnn solution (ratio $\approx 0.99$) and the spectral bound (ratio $\approx 1.25$). This confirms that the method effectively confines the true energy, providing a reliable optimality certificate.

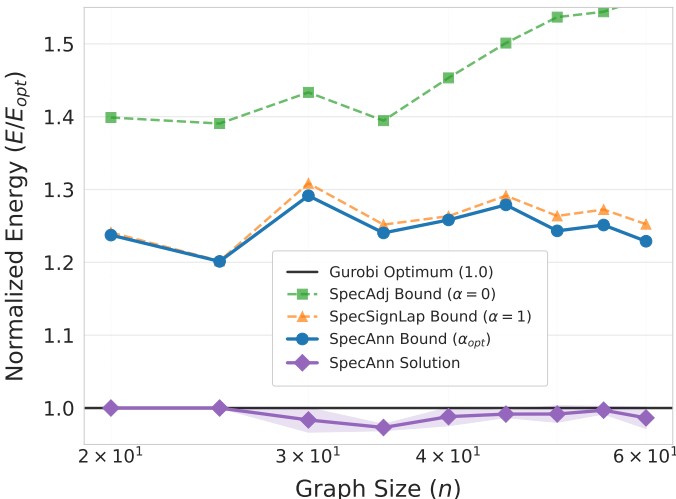

Figure 7: Tightness of spectral bounds relative to the exact ground truth ($E_{Gurobi}$) for tiny Barabási-Albert graphs.

In summary, the universal bound offers an interpretable certificate that links spectral relaxation quality with actual solution optimality. Because the bound always lies above the true energy and below the relaxed spectral estimate, it provides a transparent interval that quantifies how far any spectral method may be from the optimum.

## C.6 ALPHA SWEEP SCHEDULING

A central feature of SpecAnn is its *alpha sweeping* strategy, in which the solver iterates over different values of $\alpha$ to solve a sequence of related eigenproblems. The way in which this sweep is carried out, both in terms of the range of values considered and the number of points sampled, has a direct impact on the trade-off between runtime and solution quality.

This analysis builds on the observation in Figure 1, where the energy gap to best solution tends to reach its lowest values within the interval $\alpha \in [0, 1]$, with particularly favorable results concentrated near $\alpha = 1$. This indicates that both the chosen range and the density of sampling in that region can strongly affect performance. As a result, we examine a set of scheduling schemes that differ in interval coverage (e.g., $[0, 1]$, $[0, 1.25]$, $[0, 2]$), resolution (number of alphas sampled), and distribution (linear vs. square-root spacing). The objective is to determine which configurations yield the best compromise between energy gap reduction and computational efficiency.

Each configuration is characterized by its average gap to best solution, runtime, sweep interval, and number of sampled alphas. Table 7 summarizes the results for *square-root scaling* and *linear scaling*, indicating that square-root

scaling output better results than linear scaling in terms of solution quality and runtime. Table 8 shows that performance of interval $[0, 1]$ is twice as fast as that of $[0, 2]$, while yielding approximately the same solution quality. In table 9, the results confirm that increasing the number of alphas generally improves solution quality, but at the cost of higher runtime. Conversely, reducing the number of alphas accelerates computation but produces larger energy gaps. The comparison between square-root and linear scaling further highlights that the distribution of sampling points also matters: concentrating more values near $\alpha = 1$ can provide noticeable improvements in gap reduction without dramatically increasing runtime.

Based on this study, the sweeping schedule selected for SpecAnn is a square-root scaling over the interval $[0, 1]$ with 128 alphas. This configuration provides a balanced compromise between solution quality and runtime.

Table 7: Average gap and time for square-root vs linear scaling

|  | Sqrt scaling | Linear scaling |
| --- | --- | --- |
| Average gap (%) | **0.30** | 0.42 |
| Average time (s) | **2.31** | 2.35 |

Table 8: Average gap and time for different intervals using square-root scaling

|  | [0,1] | [0,1.25] | [0,2] |
| --- | --- | --- | --- |
| Average gap (%) | 0.29 | 0.34 | **0.28** |
| Average time (s) | **1.90** | 2.29 | 3.56 |

Table 9: Average gap and time using sqrt scaling and interval $[0, 1]$ for different alphas

|  | 21 | 32 | 51 | 64 | 101 | 128 | 201 | 256 |
| --- | --- | --- | --- | --- | --- | --- | --- | --- |
| Gap (%) | 0.54 | 0.36 | 0.35 | 0.30 | 0.26 | 0.23 | 0.16 | 0.13 |
| Time (s) | 0.48 | 0.65 | 1.015 | 1.19 | 1.88 | 2.27 | 3.52 | 4.27 |

### C.7 WARM-START STUDY

To examine the practical impact of warm-starting on the eigensolver, we measure the average number of Arnoldi iterations required at each $\alpha$-step for each instance. For this study, we evaluate SpecAnn under identical hyperparameter configurations, using two initialization strategies corresponding to the predictor variants described below:

- SpecAnn[-WS]: no warm start is used; each eigensolve begins from a random unit vector.
- SpecAnn: standard warm start diagonal predictor.
- SpecAnn[Sec]: secant predictor (Argyros & Khattri, 2013; Allgower & Georg, 1993), which interpolates from the last two eigenvectors:

$$\tilde{\mathbf{y}} = \mathbf{y}_{k-1} + \gamma(\mathbf{y}_{k-1} - \mathbf{y}_{k-2}), \qquad \mathbf{y}^{\text{init}} = \frac{\tilde{\mathbf{y}}}{\|\tilde{\mathbf{y}}\|_2},$$

with a damping parameter $\gamma \in (0, 2)$ and sign alignment applied when $\mathbf{y}_{k-1}^\top \mathbf{y}_{k-2} < 0$.

Table 10 reports the iteration count averaged across all $\alpha$-values in the square-root schedule (excluding $\alpha = 0$). Across all graph sizes, both warm-start predictors substantially reduce the number of Lanczos iterations compared with SpecAnn[-WS]. The diagonal predictor (SpecAnn) typically cuts the iteration count by 25–40%, while the secant predictor (SpecAnn[Sec]) yields the most consistent improvement, reducing iterations by 30–50% for medium and large graphs. The gap between SpecAnn and SpecAnn[Sec] grows with $n$, indicating that eigenvector drift is better captured by the two-step interpolation. These results confirm the theoretical expectation that warm-starting keeps successive eigenproblems close, and they show that predictor quality directly translates into computational savings.

Table 10: Average Lanczos iterations across BA instances.

| Size ($n$) | SpecAnn`[-WS]` | SpecAnn | SpecAnn`[Sec]` |
|---|---|---|---|
| 64 | 37.78 | **33.00** | **33.00** |
| 128 | 60.22 | **33.00** | **33.00** |
| 256 | 62.25 | 35.66 | **33.00** |
| 512 | 60.38 | 38.83 | **33.82** |
| 1024 | 64.39 | 43.69 | **35.26** |
| 2048 | 65.19 | 44.65 | **35.97** |
| 4096 | 64.85 | 43.94 | **36.97** |
| 8192 | 67.29 | 45.71 | **39.34** |
| 16384 | 69.88 | 46.87 | **40.97** |
| 32768 | 81.18 | 50.39 | **42.18** |
| 65536 | 75.52 | 48.02 | **42.67** |
| 131072 | 90.24 | 53.79 | **47.26** |
| 262144 | 103.48 | 53.94 | **49.49** |
| 524288 | 103.91 | 57.02 | **53.82** |
| 1048576 | 110.77 | **60.13** | 60.37 |

In Table 10 the observed iteration counts exhibit a lower plateau around 33 iterations. This effect is not due to a limitation of the predictors, but rather a consequence of the eigensolver design. The CuPy implementation of `eigsh` uses a *thick-restart Lanczos method* (Wu & Simon, 2000), which maintains a Krylov subspace of fixed dimension $n_{\mathrm{cv}}$ (by default $n_{\mathrm{cv}} = 33$ when $k = 1$). As a result, even when the warm start provides an initial vector extremely close to the true eigenvector, the solver must still generate $n_{\mathrm{cv}}$ Lanczos vectors before it is allowed to restart and check the convergence. In other words, SpecAnn and SpecAnn`[Sec]` already operate near the minimum iterations possible, and further reductions would require modifying $n_{\mathrm{cv}}$ or switching to a non-restarted eigensolver.

Table 2 provides an aggregate view of solution quality and runtime across the BA dataset. The differences in energy gap between the variants are negligible, indicating that warm-start strategies do not impact the final solution quality. However, the runtime results reinforce the findings of Table 10: no using warm start (SpecAnn`[-WS]`) results, on average, in 1.2 additional seconds of computation. This overhead becomes larger as $n$ grows, reaching more than 6 seconds for instances with over one million nodes.

C.8    NUMERICAL PRECISION STUDY (FP32 VS FP64)

To assess whether numerical precision influences the spectral relaxation method, we compare single precision (FP32) and double precision (FP64). For this study, we run SpecAnn on graphs ranging from $n = 2^6$ to $n = 2^{20}$ (BA). Both precisions use identical solver configurations, including the same $\alpha$-schedule. Lower-precision formats beyond FP32 are not considered here due to libraries incompatibilities.

Overall, the results show that numerical precision has only a marginal influence on the solution quality. Across all sizes, the energy gaps produced by FP32 and FP64 differ by less than $0.15\%$, with most discrepancies below $0.05\%$. Runtime differences follow the expected trend: FP64 precision implies some moderate overhead, ranging from 10–$40\%$ depending on problem size. Importantly, this overhead does not yield any systematic improvement in solution quality. These observations confirm that FP32 is sufficiently stable and FP64 offers no practical advantage.

Table 11: Comparison between SpecAnn (FP32) and SpecAnn[64] on energy gap and runtime.

| Size | Gap to Best Sol. (%) | | Time (s) | |
|---|---|---|---|---|
| | SpecAnn | SpecAnn[64] | SpecAnn | SpecAnn[64] |
| 64 | **0.00** | **0.00** | 1.60 | **1.48** |
| 128 | **0.00** | 0.04 | 1.66 | **1.49** |
| 256 | **0.00** | 0.07 | 1.80 | **1.64** |
| 512 | **0.00** | 0.15 | 1.97 | **1.79** |
| 1024 | 0.10 | **0.00** | 2.17 | **2.03** |
| 2048 | **0.00** | 0.00 | 2.20 | **2.13** |
| 4096 | 0.02 | **0.00** | 2.20 | **2.10** |
| 8192 | 0.07 | **0.00** | 2.30 | **2.28** |
| 16384 | **0.00** | 0.03 | **2.35** | 2.46 |
| 32768 | **0.00** | 0.02 | **2.57** | 2.82 |
| 65536 | **0.00** | 0.02 | **2.59** | 2.89 |
| 131072 | **0.00** | 0.04 | **3.23** | 3.79 |
| 262144 | **0.00** | 0.01 | **4.07** | 5.49 |
| 524288 | **0.00** | 0.00 | **7.45** | 10.26 |
| 1048576 | **0.00** | 0.00 | **14.89** | 20.85 |

