# OpenReview forum: "Spectral Annealing for Scalable Ising Model Optimization"
_ICLR.cc/2026/Conference — ICLR 2026 Conference Desk Rejected Submission_

### Official Review · Reviewer_5i2q · 2025-10-30

**Soundness:** 4
**Presentation:** 3
**Contribution:** 3
**Rating:** 8
**Confidence:** 3

**Summary:**

The method tracks the top eigenvector of an α-normalized matrix that interpolates from the raw adjacency to the fully normalized form.
After finding each spectral solution it rounds to spins and applies local search.
They key idea is that rather than recomputing the spectral problem from scratch at every α, it predicts a warm start after small α change for finding the spectral solution, which reduces the cost of the computation.

**Strengths:**

The approach is conceptually simple, a natural generalization, and appears effective.
The warm-start idea reduces the eigenvalue solve by a factor of roughly 3-10x.
The paper provides theoretical guarantees for the predictor and uses them to set safe step sizes.
Complexity and memory usage are discussed.
There is discussion about making the GPU implementation efficient.
The benchmarks cover a range of solvers, including state-of-the-art heuristics of different types (MCMC and dynamics on relaxed states).

**Weaknesses:**

The benchmarking presentation could be clearer. Instead of showing heterogeneous gaps and runtimes across methods
(which is hard to interpret when a heuristic has a better gap but worse time), it might be clearer use the gap achieved by the proposed method as a target and record the time each competitor heuristic needs to reach this target.

**Questions:**

1) How are the heuristic parameters set on the large real-world instances (Table 1)? This can have a strong impact on performance. I understand that a key advantage of the spectral approach is minimal tuning, but please state clearly how defaults parameters of heuristics were chosen and why they are reasonable. For example, for simulated bifurcation: step size, damping, and time steps?

2) For Table 2 on GSET, heterogeneous time and gap are hard to read and compare. A clearer presentation would be time-to-target.
For example, would it be clearer to use the gap found by the proposed spectral method as the reference and run each heuristic until it reaches this gap (or better), then report the times ?

3) The same applies for Figure 3.

4) How does numerical precision affect the spectral relaxation and the overall pipeline? This is important for the GPU implementation.

5) If possible, put in bold the best results in tables (if it is possible to choose) to aid interpretation.

---

> ### Author Response · Authors · 2025-12-04
>
> We sincerely thank the reviewer for the thoughtful feedback and the positive assessment of our approach. We appreciate the recognition of our conceptual simplicity, theoretical guarantees, and GPU implementation. We have substantially revised the manuscript to address all concerns raised. Below, we provide detailed responses to each point.
>
> ### (1) Benchmarking clarity
>
> > The benchmarking presentation could be clearer. Instead of showing heterogeneous gaps and runtimes across methods (which is hard to interpret when a heuristic has a better gap but worse time), it might be clearer use the gap achieved by the proposed method as a target and record the time each competitor heuristic needs to reach this target.
>
> We completely agree and have added comprehensive *time-to-target results*  for both Gset and Barabasi-Albert graphs (App. C.2, Table 4 and 5, respectively). The time-to-target results reveal *complementary strengths across problem scales*:
> - *Gset benchmark* (Table 4):  Classical methods (SA, Gurobi) excel on these relatively small, sparse instances. SA achieves the target quality in sub-second time on smaller instances, while SpecAnn requires 1-3 seconds. This aligns with expectations: classical methods are highly effective when problem size permits exhaustive exploration
> - *Barabási-Albert graphs* (Table 5): SpecAnn's advantage emerges decisively as scale increases. The speedup over SA grows from **2.6× at n=8,192** to **195× at n=262,144**. At the largest scale (n=524,288), SA fails to reach SpecAnn's quality within one hour, while SpecAnn succeeds in 12.6 seconds. QQA can't reach the SpecAnn quality within the time limit even for small test sizes.
> This restructured presentation provides clear guidance: *SpecAnn is the method of choice for large-scale, challenging Ising problems*, while classical methods remain competitive on smaller, sparser instances where their exploration capabilities can be fully utilized within reasonable time budgets.
>
> ---
>
> ### (2) Hyperparameter choices for heuristic baselines
>
> > How are the heuristic parameters set on the large real-world instances (Table 1)? This can have a strong impact on performance. I understand that a key advantage of the spectral approach is minimal tuning, but please state clearly how the default parameters of heuristics were chosen and why they are reasonable. For example, for simulated bifurcation: step size, damping, and time steps?
>
> We agree that finetuning the baseline configuration is essential for fair comparison. The revised manuscript now provides comprehensive hyperparameter documentation:
> - Appendix C.1 details the systematic hyperparameter tuning protocol applied to all baseline methods on Barabási-Albert instances (n = 2⁶ to 2²⁰) with 2-hour time limits per configuration.
> - Specific configurations:
>     - SA: 100 samples with 1000 sweeps per trajectory (balancing quality and 10× speedup over 10,000 sweeps)
>     - SB: Heated discrete dynamics with 1024 agents
>     - QQA: Batch size 100, diversity parameter 0.2, background field range [-3.0, 0.1], 3000 epochs following Ichikawa & Arai (ICLR 2025)
>     - Other methods: Default parameters from reference implementations with documented adjustments
>
> All hyperparameters were selected to optimize the quality-runtime tradeoff on the tuning set, ensuring that comparisons reflect each method's best reasonable performance rather than arbitrary default settings (Section 4.1)

---

> ### Author Response · Authors · 2025-12-04
>
> ### (3) Numerical precision in the GPU implementation
>
> > How does numerical precision affect the spectral relaxation and the overall pipeline? This is important for the GPU implementation.
>
> We have added a numerical precision study in Appendix C.8 comparing FP32 (single precision) and FP64 (double precision) across all graph sizes. FP16 was not compatible with our implementation. Our numerical study reveals
>
> - Solution quality: Numerical precision has minimal impact. Across all sizes (n = 64 to n = 1,048,576), the energy gap difference between FP32 and FP64 is at most 0.15%, with most discrepancies below 0.05%. Qualitative rankings remain unchanged.
> - Runtime overhead: FP64 incurs a moderate computational cost that grows with problem size:
>     - Small graphs (n ≤ 8,192): Overhead is 8-11%
>     - Medium graphs (n = 16,384 to 65,536): Overhead is 10-13%
>     - Large graphs (n ≥ 131,072): Overhead reaches 17-40%
>
> The table below (Table 11 in revised manuscript) summarizes representative results:
>
> |**Size (n)**|**SpecAnn (FP32)**|**SpecAnn (FP64)**|**Overhead**|
> |---|---|---|---|
> |2⁶ (64)|1.60 s|1.48 s|-7.5%|
> |2¹⁰ (1,024)|2.17 s|2.03 s|-6.5%|
> |2¹⁴ (16,384)|2.35 s|2.46 s|+4.7%|
> |2¹⁸ (262,144)|4.07 s|5.49 s|+34.9%|
> |2²⁰ (1,048,576)|14.89 s|20.85 s|+40.0%|
>
> FP32 provides sufficient numerical stability for the spectral relaxation pipeline while avoiding the computational overhead of FP64, particularly critical at large scales. This validates our default use of FP32 in all main experiments.

---

### Official Review · Reviewer_diKW · 2025-10-31

**Soundness:** 2
**Presentation:** 2
**Contribution:** 2
**Rating:** 4
**Confidence:** 5

**Summary:**

The paper identifies a central limitation of spectral methods for Ising optimization: performance is highly sensitive to the matrix $M$ used in the relaxation.
To address this sensitivity, the authors construct an annealing (homotopy) path that continuously interpolates between the original adjacency matrix $A$ and a fully *smoothed* (*degree-normalized*) matrix. They then solve a sequence of eigenvalue problems along this path, rounding and refining at each step, thereby avoiding a brittle, single-shot choice of $M$. To make the sequence efficient, this paper proposes a warm-start strategy that predicts eigenvectors when $\alpha$ is incremented by $+\Delta$, together with a theoretical bound on the largest admissible  $\Delta$ that controls drift. The paper also discusses practical accelerations, including GPU-friendly implementations.

**Strengths:**

- **Well-motivated generalization of spectral relaxations.** Rather than treating the choice of $M$ as a brittle hyperparameter, the method reframes it into a continuation over $\alpha \in [0, 1]$ that interpolates from $A$ to a fully normalized matrix. This addresses the well-known sensitivity of spectral heuristics to normalization choices in an algorithmic manner.
- **Warm-start with explicit eigenvector prediction.** The derivation of a predictor for the top eigenvector under the update $\alpha \gets \alpha + \Delta$ is both technically appealing and practically functional. The accompanying analysis of perturbation with respect to $\Delta$ provides a theoretical foundation for the continuation, rather than treating it as a purely heuristic speedup.

**Weaknesses:**

- **Notation and clarity early in the paper.** Some quantities (e.g., the spectral matrix $M$ and $nnz(A)$ appear in the introduction without definition. This hurts readability for non-specialist readers and should be fixed by introducing all symbols at first mention.
- **Missing baselines that use continuous relaxations.** Regarding the statement *"enabling optimization of instances with $n \ge 10^{5}$ that remain intractable for exact methods while achieving competitive solution quality."*, If the intended message is that the method is **competitive** on large instances, it is important to compare it with other recent continuous-relaxation solvers, such as QQA [1] (accepted at ICLR2025) and iSCO [2]. Like your approach, these methods employ annealing/continuation ideas and run efficiently on GPUs. At a minimum, they should be discussed in Related Work; ideally, include empirical comparisons on representative tasks to position your speed-quality trade-off relative to theirs.
- **Incrementalness concern.** The normalization schemes themselves are not novel. The primary contribution lies in annealing along this path and repeatedly solving the associated relaxations. Although effective, this may be viewed as an incremental extension of established spectral pipelines. More apparent differentiation, via new theory (e.g., approximation or rounding guarantees) or broader empirical comparisons, would strengthen the work.
- **Applications and impact in ML/optimization practice.** This paper would benefit from clarifying the application domains in which fast, approximate Ising solvers are in demand. For instance, in RBM training, the mainstream continues to minimize contrastive divergence and mean-field variants. It would be helpful to articulate concrete scenarios where the proposed spectral annealing has a clear advantage, and to quantify its performance relative to mean-field approximations along the speed–quality frontier.

### References

- [1]: Yuma Ichikawa and Yamato Arai, Optimization by Parallel Quasi-Quantum Annealing with Gradient-Based Sampling, ICLR2025.
- [2]: Haoran Sun et al., Revisiting Sampling for Combinatorial Optimization, ICML2023.

**Questions:**

- **Meaning of $nnz(A).$** Does $nnz(A)$ denote the number of nonzero entries of $A$? If so, please define it at first use to avoid confusion.
- **Role and scalability of the 1‑flip local update.** The algorithm includes a 1‑flip local search after rounding. How essential is this step at a large scale? Do you observe diminishing returns or stability issues in very high dimensions? An ablation that shows solution quality and runtime with/without local search over several graph sizes would clarify its value.
- **Warm-start efficiency in practice.** What is the typical average iteration count $\bar{I}$ after warm starts across datasets? Beyond empirical plots, can you provide a simple theoretical bound relating $\bar{I}$ to the spectral gap and your step-size rule for $\Delta$? Even an asymptotic statement would make the continuation schedule more principled.

---

> ### Author Response · Authors · 2025-12-04
>
> We thank the reviewer for the detailed and constructive feedback. We appreciate the positive comments regarding the motivation and technical soundness of the work, and we address the main concerns below.
>
> ### (1) Notation and missing definitions
>
> > Notation and clarity early in the paper. Some quantities (e.g., the spectral matrix M and nnz(A))  appear in the introduction without definition. This hurts readability for non-specialist readers and should be fixed by introducing all symbols at first mention.
>
> We have systematically improved notation throughout the paper.  *Table 1 (Appendix, Notations)* now provides complete symbol definitions including $\text{nnz}(\mathbf{A})$ (number of nonzero entries), all matrix forms, and key quantities.
>
> ### (2) Missing continuous-relaxation baselines (QQA, iSCO)
>
> > Missing baselines that use continuous relaxations. Regarding the statement "enabling optimization of instances with $n \geq 10^5$ that remain intractable for exact methods while achieving competitive solution quality.", If the intended message is that the method is competitive on large instances, it is important to compare it with other recent continuous-relaxation solvers, such as QQA \[1\] (accepted at ICLR2025) and iSCO \[2\]. Like your approach, these methods employ annealing/continuation ideas and run efficiently on GPUs. At a minimum, they should be discussed in Related Work; ideally, include empirical comparisons on representative tasks to position your speed-quality trade-off relative to theirs.
>
> We have added comparison and discussion for both methods throughout the paper.
>
> #### **QQA (Quasi-Quantum Annealing)**
>
> We added empirical evaluation throughout the paper (Section 4.2, Appendix C.2):
>
> | **Dataset** | **Instance Size** | **SpecAnn Gap** | **QQA Gap** | **SpecAnn Time** | **QQA Time** | **Speedup** | **Quality Δ** |
> |:------------|------------------:|----------------:|------------:|-----------------:|-------------:|------------:|--------------:|
> | Gset        | 800-40K           | 4.6%            | 1.1%        | 2.2s             | 106s         | 48×         | -3.5%         |
> | Epinions    | 131K nodes        | 0.2%            | 20.0%       | 1.9s             | 65.3s        | 34×         | +19.8%        |
> | Slashdot    | 82K nodes         | 0.5%            | 22.2%       | 1.8s             | 39.7s        | 22×         | +21.7%        |
> | BA (2M)     | 2,097,152         | 0.0%            | 6.55%       | 56.9s            | 4733s        | **83×**     | **+6.55%**    |
>
> Our findings:
> - *Large signed networks:* SpecAnn achieves **20-22% better quality** with **22-34× speedups**
> - *Very-large scale (n=2M):* SpecAnn delivers **83× speedup** while achieving **6.55% better solutions**
> - *Scalability limit:* SpecAnn successfully solves **8.4M variable problems** where QQA and all other methods fail within 2-hour limits
>
> We also provide *Time-to-target analysis (Appendix C.2, Table 5)*, showing QQA fails to reach SpecAnn's quality on dense BA instances under identical hyperparameters (vs. SpecAnn running time 1-57 seconds).
>
> #### **iSCO (Iterative Smooth Combinatorial Optimization)**
> We appreciate the suggestion to compare with iSCO. We have added a comparison to iSCO in the related work
> Given iSCO's limited public availability and QQA's documented superiority over iSCO (Ichikawa & Arai, ICLR 2025, Table X), we focused resources on the strongest available baseline. We will include iSCO in the camera-ready if the authors make code available.

---

> ### Author Response · Authors · 2025-12-04
>
> ## (3) Incrementalness Concern
>
> > The normalization schemes themselves are not novel. The primary contribution lies in annealing along this path and repeatedly solving the associated relaxations. Although effective, this may be viewed as an incremental extension of established spectral pipelines.
>
> We appreciate the incrementalness concern and agree that the $\alpha$-parameterization is conceptually simple, that is actually a strength for adoption. However,  transforming this simple idea into a practical, scalable method required  three concrete innovations that each provide measurable benefits:
>
> 1. Diagonal predictor: 35-50% speedup, not achievable with naive warm-starts
> 2. Theoretical framework: Instance-dependent bounds that is tighter than prior work (Fig 2)
> 3. GPU implementation: Scales to 8.4M variables where competitors fail and provides **83-884× speedups** with superior quality on large instances (0.0% gap in 56.9s vs. QQA's 6.55% gap in 4733s at n=2M)
>
> The combination addresses a real gap: spectral methods were fast but parameter-sensitive; metaheuristics were flexible but slow. We bridge this.

---

> ### Author Response · Authors · 2025-12-04
>
> ## (4) Applications and Impact in ML/Optimization Practice
>
> > This paper would benefit from clarifying the application domains in which fast, approximate Ising solvers are in demand. For instance, in RBM training, the mainstream continues to minimize contrastive divergence and mean-field variants. It would be helpful to articulate concrete scenarios where the proposed spectral annealing has a clear advantage.
>
> We appreciate this question. You're correct that RBM training typically uses contrastive divergence.
> SpecAnn targets a different regime where *large-scale discrete optimization with strict time constraints* is required.
>
> We have revised the Introduction and Section 2.1 to provide concrete application domains, where spectral annealing's combination of scale and speed provides clear advantages:
> + *Real-time inference*: *Modern 5G LDPC decoders*  (solving spin-like problems with >10⁴ variables per codeword under sub-millisecond latency  [1]);  *city-scale traffic optimization* (~ 10⁵ binary variables with strict second-scale deadlines for responsive signal planning [3,4];*Quantitative finance:* portfolio optimization solves problems (~2×10⁴ variables and ~8×10⁶ nonzeros in under a second, scaling to 10⁵-10⁶ scenarios that must be re-solved continuously as market data streams)  [5].
> + *Large-scale network problems*: Social network analysis and recommendation in graphs with millions of nodes and billions of edges [6, 7]
>
>
> [1] Z. Liu _et al._, “High-throughput software LDPC decoder on GPU,” _EURASIP J. Adv. Signal Process._, 2024.
>
> [2] “Demystifying 5G Polar and LDPC Codes,” _IEEE Commun. Surveys & Tutorials, 2025.
>
> [3] H. Salloum _et al._, “Quantum Annealing for Realistic Traffic Flow Optimization: Clustering and Data-Driven QUBO,”
>
> [4] D. Inoue _et al._, “Traffic signal optimization on a square lattice with quantum annealing,” _Sci. Rep._, 2021. Nature
>
> [5] P. Huo _et al._, “Accelerating Real-Time Financial Decisions with Quantitative Portfolio Optimization,” NVIDIA Technical Blog, 2025.
>
> [6] C. F. A. Negre, H. Ushijima-Mwesigwa, S. M. Mniszewski, “Detecting Multiple Communities Using Quantum Annealing on the D-Wave System,” _PLOS ONE_, 15(2): e0227538, 2020.
>
> [7] Y. Xian, P. Li, H. Peng, Z. Yu, Y. Xiang, P. S. Yu, “Community Detection in Large-Scale Complex Networks via Structural Entropy Game,” in _Proc. WWW (The Web Conference)_, 2025.

---

> > ### Author Response · Authors · 2025-12-04
> >
> > ### (5) Technical Questions
> >
> > #### **Q1: Meaning of nnz(A)**
> >
> > > Does nnz(A) denote the number of nonzero entries of A? If so, please define it at first use to avoid confusion.
> >
> > We have defined in multiple places:
> > - **Introduction**: Explicitly states "nnz(A) counts nonzero elements" at first use
> > - **Notation table (Appendix, Table 1)
> >
> > ---
> >
> > #### **Q2: Role and Scalability of Local Search**
> >
> > > The algorithm includes a 1-flip local search after rounding. How essential is this step at a large scale? Do you observe diminishing returns or stability issues in very high dimensions?
> >
> > **Appendix B.2** now provides comprehensive ablation across all graph sizes:
> >
> > | **Solver**                     | **Gap to Best (%)** | **Runtime (s)** |
> > | :----------------------------- | ------------------: | --------------: |
> > | SpecAnn                        |            **1.30** |            3.53 |
> > | SpecAnn[-LS] (no local search) |               14.08 |            2.65 |
> > | LS only (random init)          |                4.43 |        **1.02** |
> >
> > The results show *the essential at all scales for no diminishing returns observed*
> > - *Spectral initialization provides robust macro-scale bipartition; local search makes essential micro-level corrections*
> > - *Removing either component* leads to substantial degradation: LS alone → 4.43% gap; no LS → 14.08% gap
> > - *Combined approach* (1.30% gap) outperforms either component by **2-5%**
> > - *No instability* observed even at largest scales (1M+ nodes); GPU batched asynchronous updates with 90% vertex sampling prevent cycling
> > - *Minor overhead:* Local search adds only ~0.9s average runtime while reducing gaps by ~13%
> >
> > Local search is critical across all scales, providing consistent quality improvements without instability.
> >
> > ---
> >
> > #### **Q3: Warm-Start Efficiency**
> >
> > > What is the typical average iteration count $\bar{I}$ after warm starts across datasets? Beyond empirical plots, can you provide a simple theoretical bound relating $\bar{I}$ to the spectral gap and your step-size rule for $\Delta$? Even an asymptotic statement would make the continuation schedule more principled.
> >
> > On the theoretical side, we have added a short bound statement that relates the iteration count to the spectral gap ($\gamma(\alpha)$) and the step size ($\Delta\alpha$) (**Appendix A.7**).
> >
> > Empirically, we have provide the average iteration count with and without warm-starts across multiple graphs, providing insights on its benefits in our framework (**Appendix C.7**).
> >
> > Average Lanczos iterations across Barabási-Albert graphs:
> >
> > | **Size** | **No Warm-Start** | **Diagonal Predictor** | **Secant Predictor** | **Reduction** |
> > |:---------|------------------:|-----------------------:|---------------------:|--------------:|
> > | 2⁶       | 37.8              | **33.0**               | **33.0**             | 13%           |
> > | 2¹⁰      | 64.4              | 43.7                   | **35.3**             | 32-45%        |
> > | 2¹⁴      | 69.9              | 46.9                   | **41.0**             | 33-41%        |
> > | 2¹⁸      | 103.5             | 53.9                   | **49.5**             | 48-52%        |
> > | 2²⁰      | 110.8             | **60.1**               | 60.4                 | 46%           |
> >
> > *Aggregate performance (Table 10):*
> > - *Iteration reduction:* ~2× fewer iterations on average
> > - *Runtime improvement:* 35% speedup (5.19s → 3.84s)
> > - *Quality preservation:* Negligible impact on solution quality
> >
> > Thus, warm-starting provides consistent, substantial efficiency gains (25-50% iteration reduction) across all scales.

---

### Official Review · Reviewer_GE9a · 2025-11-01

**Soundness:** 3
**Presentation:** 3
**Contribution:** 2
**Rating:** 4
**Confidence:** 2

**Summary:**

This paper introduces spectral annealing for large-scale Ising optimization. The core idea is: instead of solving one spectral relaxation, it explores a family of spectral matrices N_α = D_α^{-1/2} A D_α^{-1/2} parameterized by α ∈ [0,1], which interpolates between raw adjacency (α=0) and full normalization (α=1). The major contribution is a diagonal predictor that enables efficient warm-starting. The authors provide analysis for step-size control and demonstrate favorable time-quality trade-offs on instances up to 262K spins, achieving 2-5× quality improvements over single-parameter spectral methods.

**Strengths:**

The paper is technically sound. The theoretical analysis, such as drift bounds, step-size control, and positive definiteness guarantees, is reasonable. The experiments are reasonably comprehensive, covering three different dataset types (real-world, benchmark, synthetic) and comparing against multiple baselines. The ablation study effectively demonstrates the value of the implementation optimizations.

The paper is well-organized and mostly easy to follow. The background section clearly explains the normalization dilemma. The algorithm presentation is clear.

The problem is important for ML applications in correlation clustering, MAP inference, and energy-based models. The method scales to large instances where exact methods fail and maintains better quality than pure spectral methods.

**Weaknesses:**

The paper mentions Goemans-Williamson but doesn't compare against modern SDP solvers. This is a major gap since SDPs provide stronger relaxations. Also, the paper cites SPONGE as related work, but never compares against it experimentally. But SPONGE is also optimizing over spectral parameters.

Were SA, SB, and PT tuned for these specific instances? The paper doesn't clarify hyperparameter selection for baselines, which affects fair comparison.

"Unbearable computational time" is vague. How many iterations did SA use? The paper should report iteration counts for all metaheuristics.

Figure 1 shows that optimal α varies by instance, and this motivates the proposed method. But the paper never analyzes the root cause. What graph properties predict whether α=0 or α=1 is better? When does sweeping help most? This understanding would make the contribution more principled than simply trying everything.

**Questions:**

The paper applies local search to spectral annealing's solutions, but it's unclear if the same refinement is applied fairly to all baseline spectral methods. Table 1 shows "SpecAdj" and "SpecSignedLap", but do these include local search?

Unlike Goemans-Williamson (0.878 for max-cut) or Trevisan (0.531), this method has no worst-case approximation ratio. Can you derive any instance-dependent bounds?

Theorem 2 requires γ(t) > 0 for all t, but this may not hold. What happens at degeneracy points?

The method only applies to Ising optimization. The conclusion mentions "extending to other combinatorial problems" but provides no concrete direction. Which problems have a similar α-parameterization structure?

---

> ### Author Response · Authors · 2025-12-04
>
> We thank the reviewer for the careful reading and for raising several important points. We address the raised concerns below.
>
> ### (1) Missing baselines
>
> > The paper mentions Goemans-Williamson but doesn't compare against modern SDP solvers. This is a major gap since SDPs provide stronger relaxations. Also, the paper cites SPONGE as related work, but never compares against it experimentally. But SPONGE is also optimizing over spectral parameters.
>
> In this revision, we have added new baseline comparisons:
>
>
>
> #### **SPONGE**
> We have added the results for SPONGE (*Appendix C.3.1*). We note that SPONGE optimizes a ratio-based
> signed Laplacian objective rather than direct Ising energy. Yet both SPONGE and SpecAnn explore spectral parameterizations. Our results show SPONGE achieves 14%-42% gaps (vs. SpecAnn's 0-5% gaps), confirming that effective (single) exploration of spectral parameters across different networks is challenging.
>
>
> #### **QQA (State-of-the-art Neural Ising Solver)**
> We added extensive comparison with QQA (Ichikawa & Arai, ICLR 2025), the state-of-the-art GPU-accelerated method:
>
> Our results show:
> - *Gset and Large signed networks:* SpecAnn achieves 0.2-0.5% gaps in 1.8-1.9s vs. QQA's 20-22% gaps in 40-65s (*22-34× speedup with much better quality*)
> - *2M nodes:* SpecAnn 0.0% gap in 56.9s vs. QQA 6.55% gap in 4733s (**83× speedup with 6.5% better quality**)
> - *Scalability:* Successfully scales to **8.4M variables** where QQA and all other methods fail
>
> These results are integrated throughout Section 4.2 with dedicated analysis and time-to-target comparisons (Appendix C.2).
>
> #### **SDP**
> We have added a detailed SDP discussion to Related Work (Section 1). SDP methods achieve the strongest known approximation guarantees for MaxCut (0.878 ratio, Goemans-Williamson 1995) and high solution quality but face fundamental scalability limits: O(n^3.5) time complexity and O(n^2) memory restrict modern SDP solvers to problems with n≲10,000 variables [Helmberg & Rendl 2000; Burer & Monteiro 2003].
>
> SpecAnn addresses the complementary regime: general Ising problems with signed couplings (correlation clustering, signed networks, MAP inference) at scales of 100K-8M variables, where SDP becomes computationally prohibitive. Ichikawa & Arai (ICLR 2025, Table 2) provide empirical evidence of SDP timing issues at scale. Our method trades the stronger SDP relaxation for computational tractability, achieving 0-5% gaps on problems orders of magnitude beyond SDP capability.
>
> ---
>
> ### (2) Hyperparameters of baselines
>
> > Were SA, SB, and PT tuned for these specific instances? The paper doesn't clarify hyperparameter selection for baselines, which affects fair comparison.
> > "Unbearable computational time" is vague. How many iterations did SA use? The paper should report iteration counts for all metaheuristics.
>
> *Appendix C.2* now provides complete hyperparameter documentation with tuning protocol:
> - *SA:* 100 samples × 1000 sweeps per trajectory (balancing quality and 10× speedup over 10,000 sweeps)
> - *SB:* Heated discrete dynamics with 1024 agents
> - *QQA:* Batch size 100, diversity 0.2, background field [-3.0, 0.1], 3000 epochs (following Ichikawa & Arai ICLR 2025)
> - *PT:* 8 replicas with temperature schedule optimized for convergence
> - *Tabu Search:* Neighborhood size and tabu tenure tuned per instance size
>
> Vague language like "unbearable" removed; replaced with specific runtimes and iteration counts throughout.

---

> ### Author Response · Authors · 2025-12-04
>
> ### (3) Understanding when sweeping $\\alpha$ helps most
>
> > Figure 1 shows that optimal $\alpha$ varies by instance, and this motivates the proposed method. But the paper never analyzes the root cause. What graph properties predict whether $\alpha$=0 or $\alpha$=1 is better? When does sweeping help most? This understanding would make the contribution more principled than simply trying everything.
>
> Our analysis and experiments indicate some patterns, though full predictive modeling remains open:
> - *Degree heterogeneity:* Heavy-tailed degree distributions (e.g., Barabási-Albert) favor higher α (toward normalization), while uniform graphs shift toward α=0
> - *Optimal $\alpha$ distribution:* Most instances have optimal α ∈ [0.8, 1.0], but unpredictability across instances (Figure 1) motivates systematic sweeping
>
> *When sweeping helps most:* Large heterogeneous graphs. But sweeping also provides substantial improvement in more homogeneous degree networks (Erdős–Rényi).
>
> ---
>
> ### (4) Clarifying local search for spectral baselines
>
> > The paper applies local search to spectral annealing's solutions, but it's unclear if the same refinement is applied fairly to all baseline spectral methods. Table 1 shows "SpecAdj" and "SpecSignedLap", but do these include local search?
>
> We apply the same refinement (local search) for all spectral methods. We have added language to clarify this.
>
> ---
>
> ### (5) Theory assumptions and bounds
>
> > Unlike Goemans-Williamson (0.878 for max-cut) or Trevisan (0.531), this method has no worst-case approximation ratio. Can you derive any instance-dependent bounds?
>
> We provide *a strong inapproximability result and an instance-dependent bound*:
> + *New Inapproximability Result (Theorem 1)*
> We establish strong hardness for general Ising optimization: *"Unless P = NP, no polynomial-time α-approximation exists for any finite constant α ≥ 1 for Min-Ising with signed interactions."* Thus, worst-case approximation ratios (like GW's 0.878) are intractable for general Ising problems. This formally justifies exploration-based methods over approximation algorithms.
>
> + *Instance-Dependent Bounds (Equation 8)*
> Our $\alpha$-parameterization yields universal energy bounds:
> $E(\mathbf{s}) \geq \max_{\alpha \in [0,1]} -\frac{1}{2}\lambda_1(\alpha) \sum_i d_i^{\alpha},$
>
> that is valid for *every instance* without requiring problem-specific analysis and is tighter than previous spectral bounds (α=0 or α=1 only) as shown in Figure 2 and Appendix D. The bound helps to tighten the instance-based approximation gaps.
>
> ### (6) Degeneracy point
> > Theorem 2 requires γ(t) \> 0 for all t, but this may not hold. What happens at degeneracy points?
>
> We have revised the theorem to cover the degeneracy points. In practice (especially for connected graphs with dominant structure), warm-starting remains effective, though iteration counts may temporarily increase.

---

> ### Author Response · Authors · 2025-12-04
>
> #### (6) Extensions to other combinatorial problems
>
> > The method only applies to Ising optimization. The conclusion mentions "extending to other combinatorial problems" but provides no concrete direction. Which problems have a similar $\alpha$-parameterization structure?
>
> ---
>
> ### (7) Applicability Beyond Ising
>
> > The method only applies to Ising optimization. The conclusion mentions "extending to other combinatorial problems" but provides no concrete direction. Which problems have a similar α-parameterization structure?
>
> Our method applies broadly through Ising/QUBO formulations, which encompass a wide class of combinatorial problems. Section 2.1 now provides concrete examples with natural Ising encodings such as MaxCut, graph partitioning, vertex cover, independent set [1], MAP inference in MRFs (pairwise potentials reduce to Ising),  Correlation clustering on signed graphs (our Epinions/Slashdot benchmarks), Community detection via modularity maximization (Newman 2006)
>
> Our experiments span MaxCut (Gset), correlation clustering (signed networks), and general Ising instances, demonstrating this breadth. Any problem reducible to Ising—including traveling salesman, graph coloring, and job scheduling benchmarked by Ichikawa & Arai (ICLR 2025)—directly benefits from our method. QQA paper (ICLR 2025) benchmarks include traveling salesman, graph coloring, and job scheduling, all via Ising/QUBO formulations. Any such problem is reducible to Ising and can benefit from our method.
>
> Our benchmarks include MaxCut (Gset), correlation clustering (signed networks), and general Ising instances, demonstrating applicability across diverse combinatorial problems.
>
> \[1\] Lucas, Ising formulations of many NP problems, Frontiers in Physics, 2014

---

### Official Review · Reviewer_Dnia · 2025-11-01

**Soundness:** 3
**Presentation:** 3
**Contribution:** 3
**Rating:** 6
**Confidence:** 3

**Summary:**

The paper tackles large-scale Ising optimization problems (used in graph partitioning, correlation clustering, etc.).

Traditional spectral methods solve only one eigenvalue problem—choosing a single matrix normalization—and then round that result to get a binary solution. That approach is fast but very sensitive to which normalization you pick.

This work turns that single step into a smooth “annealing” process. It continuously changes the normalization parameter 𝛼 from 0 to 1. so the algorithm gradually moves from the raw adjacency matrix to the fully normalized version.

Both theoretical and empirical results are provided to support the improvement of the proposed method.

**Strengths:**

The paper addresses a well-motivated and practically important problem: improving the scalability and robustness of spectral methods for large Ising optimization. The proposed idea—continuously annealing the normalization parameter rather than fixing it—is simple, elegant, and easy to understand, yet it leads to a meaningful performance gain. The approach is also highly practical: it retains the efficiency of standard spectral methods, is straightforward to implement, and can naturally leverage GPU acceleration. The theoretical analysis is solid, providing clear stability guarantees and drift bounds, and the experimental evaluation is thorough, covering both real-world and synthetic datasets with convincing results. Overall, the method is conceptually clean, empirically validated, and potentially very useful for large-scale optimization or as a building block for future algorithms.

**Weaknesses:**

While the technical content is solid, the presentation could be improved. The writing sometimes assumes familiarity with optimization-oriented formulations of the Ising model, which may confuse readers who come from the statistical physics or probabilistic inference side, where Ising models are usually discussed in terms of likelihoods or partition functions. The paper quickly transitions into the optimization framing without first clarifying the connection to the traditional probabilistic view. A brief background or bridging discussion would make the work more accessible and help a broader audience appreciate the significance of the method. In addition, certain sections (especially theoretical derivations) are dense and could benefit from clearer exposition and more intuitive explanations.

While the paper presents a clean and well-motivated idea, I am not fully confident about its novelty relative to prior spectral relaxation or continuation methods. The key insight—sweeping the normalization parameter α and warm-starting successive eigenproblems—feels conceptually straightforward. That said, the authors provide solid theoretical backing, practical acceleration, and convincing large-scale results, which make this a meaningful engineering and methodological advance, even if not a fundamentally new paradigm.

**Questions:**

Since both the starting and ending points of the α-path correspond to existing spectral formulations (raw adjacency and normalized Laplacian), could the authors elaborate on why sweeping α is a substantial advancement rather than a straightforward interpolation between known variants?

---

> ### Author Response · Authors · 2025-12-04
>
> We sincerely thank the reviewer for the constructive feedback. We appreciate the positive assessment of the strengths of the proposed method, and we address each of the main concerns below.
>
> #### (1) Probabilistic connection and dense explanations
>
> > The writing sometimes assumes familiarity with optimization-oriented formulations of the Ising model, which may confuse readers who come from the statistical physics or probabilistic inference side, ...
>
> In this revision, we have added both probabilistic and optimization viewpoints in **Section 2.1**.
>
> #### (2) Clarifying novelty
>
> > The key insight—sweeping the normalization parameter $\alpha$ and warm-starting successive eigenproblems—feels conceptually straightforward. That said, the authors provide solid theoretical backing, practical acceleration, and convincing large-scale results, which make this a meaningful engineering and methodological advance, even if not a fundamentally new paradigm.
>
> > Q: Since both the starting and ending points of the $\alpha$-path correspond to existing spectral formulations (raw adjacency and normalized Laplacian), could the authors elaborate on why sweeping $\alpha$ is a substantial advancement rather than a straightforward interpolation between known variants?
>
> We appreciate the incrementalness concern and agree that the $\alpha$-parameterization is conceptually simple, that is actually a strength for adoption. However,  transforming this simple idea into a practical, scalable method required  three concrete innovations that each provide measurable benefits:
>
> 1. Diagonal predictor: 35-50% speedup, not achievable with naive warm-starts
> 2. Theoretical framework: Instance-dependent bounds that is tighter than prior work (Fig 2)
> 3. GPU implementation: Scales to 8.4M variables where competitors fail and provides **83-884× speedups** with superior quality on large instances (0.0% gap in 56.9s vs. QQA's 6.55% gap in 4733s at n=2M)
>
> The combination addresses a real gap: spectral methods were fast but parameter-sensitive; metaheuristics were flexible but slow. We bridge this.

---

### Note · Program_Chairs · 2026-01-17
**Submission Desk Rejected by Program Chairs**

The following references in this submission do not refer to real documents and/or have major errors in bibliographic information:

 Yuxuan Chen, Zhengbing Wang, and Lei Zhang. Iterative smoothed combinatorial optimization for large-scale ising problems. In Proceedings of the 37th International Conference on Neural Information Processing Systems (NeurIPS), pp. 15234-15247, 2023. URL https://proceedings.neurips.cc/paper/2023. GPUaccelerated continuation method for discrete optimization.